# Vowel onset measures and their reliability, sensitivity and specificity: A systematic literature review

**Antonia Margarita Chacon**[1]*, **Duy Duong Nguyen**[1], **John Holik**[1], **Michael Döllinger**[2], **Tomás Arias-Vergara**[2,3], **Catherine Jeanette Madill**[1]

**1** Voice Research Laboratory/ Doctor Liang Voice Program, Discipline of Speech Pathology, Faculty of Medicine and Health, Sydney School of Health Sciences, The University of Sydney, Sydney, NSW, Australia, **2** Division of Phoniatrics and Paediatric Audiology at the Department of Otorhinolaryngology Head & Neck Surgery, University Hospital Erlangen, Friedrich-Alexander-Universität Erlangen-Nürnberg, Erlangen, Bavaria, Germany, **3** Department of Computer Science, Chair of Computer Science 5, Friedrich-Alexander-University Erlangen-Nürnberg, Erlangen, Bavaria, Germany

* antonia.chacon@sydney.edu.au

## Abstract

### Objective

To systematically evaluate the evidence for the reliability, sensitivity and specificity of existing measures of vowel-initial voice onset.

### Methods

A literature search was conducted across electronic databases for published studies (MEDLINE, EMBASE, Scopus, Web of Science, CINAHL, PubMed Central, IEEE Xplore) and grey literature (ProQuest for unpublished dissertations) measuring vowel onset. Eligibility criteria included research of any study design type or context focused on measuring human voice onset on an initial vowel. Two independent reviewers were involved at each stage of title and abstract screening, data extraction and analysis. Data extracted included measures used, their reliability, sensitivity and specificity. Risk of bias and certainty of evidence was assessed using GRADE as the data of interest was extracted.

### Results

The search retrieved 6,983 records. Titles and abstracts were screened against the inclusion criteria by two independent reviewers, with a third reviewer responsible for conflict resolution. Thirty-five papers were included in the review, which identified five categories of voice onset measurement: auditory perceptual, acoustic, aerodynamic, physiological and visual imaging. Reliability was explored in 14 papers with varied reliability ratings, while sensitivity was rarely assessed, and no assessment of specificity was conducted across any of the included records. Certainty of evidence ranged from very low to moderate with high variability in methodology and voice onset measures used.

**Data Availability Statement:** All relevant data are within the manuscript and its Supporting Information files.

**Funding:** Contributions by MD were supported by Deutsche Forschungsgemeinschaft (DFG) under grant no. DO1247/8-2. Contributions by TA were supported by Deutsche Forschungsgemeinschaft (DFG) under grant no. SCHU3441/3-2. Contributions by AC, DN and JH were supported by the Doctor Liang Voice Program at The University of Sydney. The funders had no role in study design, data collection and analysis, decision to publish, nor preparation of the manuscript.

**Competing interests:** The authors have declared that no competing interests exist.

## Conclusions

A range of vowel-initial voice onset measurements have been applied throughout the literature, however, there is a lack of evidence regarding their sensitivity, specificity and reliability in the detection and discrimination of voice onset types. Heterogeneity in study populations and methods used preclude conclusions on the most valid measures. There is a clear need for standardisation of research methodology, and for future studies to examine the practicality of these measures in research and clinical settings.

## Introduction

Measures of voicing control provide critical insight into a myriad of voice diagnoses across the lifespan. Voice disorders are highly prevalent, with an estimated one in thirteen adults experiencing a voice disorder each year [1]. Early and accurate diagnosis are essential to optimise patients' vocal health outcomes. Traditionally, voice assessment and the evaluation of voice rehabilitative outcomes have focused upon voice quality [2, 3] and patient-reported outcomes [4] as measures of voice function and efficiency. This assessment proforma typically involves the collection of a patient's case history information, acoustic voice assessment and auditory perceptual judgement of the patient's voice quality. Ideally, these tasks are also supplemented by laryngostroboscopic and aerodynamic assessment [5, 6]. Most current voice assessment methods prioritise steady-state phonation with little, if any focus placed upon the initiation of voicing. Voice onset predicts the voice function that follows and as such, has been increasingly suggested as an effective means of assessing one's voice, providing predictive information about phonation type, facilitating voice disorder diagnosis and determining one's response to treatment [7–10].

Voice onset refers to the span of time between the release of a sound and the onset of voicing and involves several physiological processes. The onset of voice begins with transglottal airflow from the lungs bypassing the larynx and the start of vocal fold adduction. Small-amplitude, irregular vibration occurs at the edges of the vocal folds bordering the open glottis. Following the first instance of medial vocal fold contact, the amplitude of these vibrations grows, and steady-state oscillations are established [11, 12]. The various physiological components involved in the onset of voice introduce many different means of voice onset measurement. There is also the compounding issue of differing types of voice onset. These are most commonly referred to as soft, breathy and hard, which are discernible to varying degrees depending on the measurement used.

There are two types of voice onset; one occurs after the release of a stop consonant and the other involves vowel phonation without a preceding consonant. Measures of voice onset which focus on the interval between the initial burst of a stop consonant and the voicing onset of the following vowel, e.g., 'Voice Onset Time' (VOT) [13], have been studied widely across populations and health statuses for many decades. The seminal papers in the voice onset literature typically relate to these such contexts of voice onset [14–18], as do most papers within the voice onset literature [12], with definitions of vowel-initial voice onset often being less clear. The onset of voicing which occurs when a vowel follows a consonant (CV), versus vowel-initial contexts of voicing varies considerably from a measurement perspective. CV measurement requires the ability to detect and differentiate between a consonant and vowel sound before analysing the vowel onset production, while vowel-initial contexts involve detection and

measurement from the very start of voicing. Vowel-initial voice onset measurement is more clinically relevant than the measurement of CV productions, as vowel production is one of the standardised tasks performed in voice assessment [19–21]. It also allows for an indication of a patient's voice production without the articulatory influences which are present in consonant-initial contexts [22]. Furthermore, the classification of voice onset types has been based primarily on vowel-centric tasks, and not upon vocal productions commencing with a consonant sound [7, 23], and yet, vowel-initial voice onset has been researched to a lesser extent than CV voicing. As such, exploring the current state of the literature for specifically vowel-initial voicing onsets has been selected as a focus for this review.

The means through which voice onset has been measured across the existing evidence base is highly variable and has evolved with technological advances over time. Researchers measure voice onset through a range of measurement types, such as auditory perceptual measures, which involve making a judgement about the properties of a sound [23–25]; aerodynamic measures, such as phonatory airflow, volume and pressure [26–28]; physiologically, which monitors the physiological muscle movement associated with voice onset [11, 29, 30]; acoustically, which examines voice signal characteristics related to speech and voice production [12, 31, 32]; visually, through high-speed laryngoscopic examination of the vocal fold vibration associated with voice onset production [33–35], or through a combination of these [36–38]. Each of these methods of voice onset measurement present their own respective strengths and weaknesses, pertaining to the ability of each measure to reflect phonatory function or account for speaker variability, the reliability, sensitivity and specificity of the resulting measurement values, and factors associated with specific equipment requirements, training or skill-level in performing each measurement type. Nonetheless, no literature yet exists which has synthesised and consolidated the measures of voice onset which have been investigated, which are the most reliable, specific and sensitive in identifying or differentiating voice onset types, the contexts in which these measures may best be used, nor established a common language amongst voice onset types and the implications of these upon vocal function. It is imperative that these research gaps be filled so that valid clinical measures of voice onset can be established, which, in turn, can facilitate the inclusion of vowel onset measurement as part of the standardised clinical voice assessment proforma. The aim of this systematic review is to evaluate the evidence for sensitivity, specificity and reliability of vowel-initial voice onset measures, with the authors hypothesizing that high reliability, sensitivity and specificity ratings will indicate the most effective measures of vowel onset. To this end, the proposed systematic review will answer the following question: What are the methods of assessing vowel-initial voice onset and the evidence for their reliability, sensitivity and specificity?

## Methods

### Protocol and registration

This retrospective systematic review was conducted according to the Preferred Reporting Items for Systematic Reviews and Meta-Analyses (PRISMA) [39]. The protocol was registered through the PROSPERO International Prospective Register for Systematic Reviews (registration number CRD42021266384) and is provided in S1 File. The completed PRISMA 2020 checklist is provided in S1 Checklist.

### Information sources

Databases searched were MEDLINE via OVID, EMBASE via OVID, Scopus, Web of Science, IEEE Xplore, CINAHL and PubMed Central. Grey literature was also searched through ProQuest to capture unpublished dissertations.

## Search strategy

The initial search was conducted by AC in August 2021 and limited to articles published after January 1900. The search strategy was initially determined through discussions between four authors (AC, CM, MD, DN). The first author also conducted an updated search in December 2022 and May 2023 to capture any further articles of relevance ahead of publication.

The search string consisted of terms relating to three 'concept areas': voice onset, voice onset measures and evidence for measures of voice onset. Within the selected concept areas, we developed a list of synonyms and/or specific terms relevant to our search scope. The terms associated with each concept area were searched against the other concept word lists to achieve literature saturation of all relevant articles. The search strategies and Boolean operators applied to the MEDLINE, EMBASE, Scopus, Web of Science, IEEE Xplore, CINAHL, PubMed Central and ProQuest databases are provided in S2 File.

## Inclusion criteria

The scope of this literature review was the onset of vowel phonation without a preceding consonant. Studies and unpublished works were included if they were written in English, related to measures of human voice onset and were published after 1900. Nil study design limits were enforced, nor were specific settings of interest; research occurring in both laboratory and clinical settings were included. Articles were excluded if they related to the onset of artificial or computerised tones, examined voice onset in vowels following the production of a consonant sound (i.e., Voice Onset Time) and/or were not written in the English language.

## Study records

The database searches retrieved 6,983 records. These records were uploaded to the Covidence platform (www.covidence.org) to manage data, facilitate collaboration and document the review process over the course of the study.

Covidence identified 550 duplicates which were then removed for a total of 6,433 records. Titles and abstracts were screened against the inclusion criteria by two independent reviewers (any combination of MD, AC, DN, JH and TA). Any disagreements which arose between the reviewers at each stage of the selection process were resolved through the involvement of a third reviewer. Five thousand, nine hundred and twenty-two records were excluded based on titles and abstracts, with a further 11 studies being excluded as their papers could not be retrieved. Full texts of the remaining 500 records were assessed in detail against the inclusion criteria by two independent reviewers (any combination of DN, AC, MD, TA, JH and CM). Articles that did not meet the study criteria were removed, with reasons for exclusion being recorded. Four hundred and seventy-two papers were excluded from this process. For the purposes of literature saturation, a further hand search of the remaining articles' citation lists was conducted (AC). Following a further process of title/abstract screening (MD, AC, DN, JH and TA), full text review and exclusion of inappropriate studies (AC, DN, MD, JH and TA), an additional seven studies were included.

An updated review of the literature was conducted in December 2022 and May 2023. The processes of title/abstract screening (AC, DN, JH), full text review and exclusion of inappropriate studies (AC, DN, JH), were again completed. The December 2022 search found nil further studies appropriate for inclusion, while the search conducted in May 2023 identified a further two studies. The final systematic review included 35 studies. A visual representation of this process is shown in Fig 1, formatted according to the PRISMA 2020 statement [39].

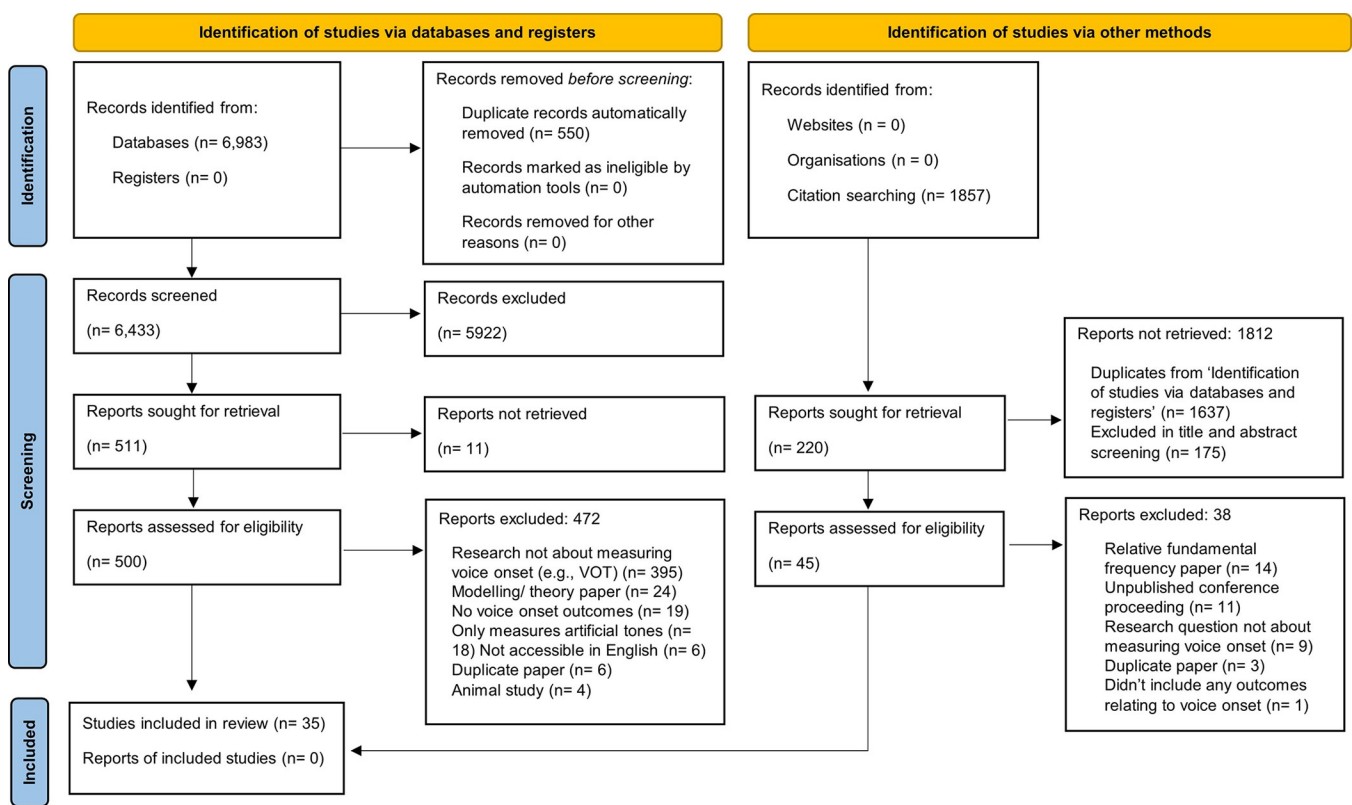

**Fig 1. PRISMA flowchart of search processes.**

## Data extraction and data items

Data was extracted from the included papers by all members of the research team. The data extraction process involved each team member reading the paper in its entirety, before extracting all information of relevance into the data extraction table. A simplified version of this table is presented in S3 File and the OSF Home Repository (DOI 10.17605/OSF.IO/N65SX). Quantitative synthesis and meta-analyses were not completed owing to the heterogeneity of data and methodologies across studies. Rather, studies were grouped according to their voice onset measurement category (see Table 1). Following the study groupings, the data extracted from all studies across each measurement category was closely examined to identify key relationships and discrepancies across and between papers and categories. This informed the key research findings which are summarised in the Results section.

## Evaluation of certainty of evidence and risk of bias

The certainty of the included evidence was assessed through the Grading of Recommendations Assessment, Development and Evaluation (GRADE) working group methodology [40]. This involved each reviewer examining the quality of evidence through the domains of risk of bias, consistency, precision, directness and publication bias. This was particularly facilitated using the GRADE Handbook [41], which was used by team members to inform their assessment and provide a consistent evaluation across raters. Following this evaluation, it was determined whether the quality of the research could be deemed as high (i.e. very unlikely that further research will change our confidence in the estimate of effect), moderate (i.e. likely that further research will have an impact on our confidence in the estimate of effect and may change the

**Table 1. Summary of individual study characteristics.**

| Study details | Study design | Setting | Characteristics of study population | Details of participants (age, gender, number) | Voice onset measurement categories explored | Certainty of evidence (GRADE) |
|---|---|---|---|---|---|---|
| Baken, & Watson (2019) [29] | Review paper with case examples | Laboratory | Not specified | Not specified, however seemed to range from <24 to >40-year-old women and men | Acoustic, physiological | Very low |
| Braunschweig, Flaschka, Schelhorn-Neise & Döllinger (2008) [31] | Solution validation with cross-sectional data | Clinical | Vocally healthy speakers and speakers with mild functional dysphonia | N = 71 females aged 18–24 years | Acoustic, visual imaging | Low |
| Choi, Oh S & Choi (2015) [33] | Cross-sectional | Laboratory | Vocally healthy speakers | N = 40 (20F, 20M) non-native Korean speakers. Females aged 20–26 years (mean 21.9); males aged 20–28 years (mean 22.7) | Visual imaging | Very low |
| Cohen, Cohen, Benyamini, Adi & Keshet (2019) [32] | Cross-sectional | Clinical | Control group: vocally healthy Patient group: unilateral VF paralysis | N = 38 (18 controls; 20 patients with unilateral VF paralysis) Controls: 10F, 8M; Patients: 8F, 12M | Acoustic | Very low |
| Cooke, Ludlow, Hallett & Selbie (1997) [7] | Cross-sectional | Laboratory | Vocally healthy speakers | N = 10 (5F, 5M). Ages range from 18–32 years (mean age 25.1 years) | Auditory perceptual, visual imaging | Very low |
| Freeman, Woo, Saxman & Murry (2012) [36] | Cross-sectional | Laboratory | Vocally healthy speakers | N = 2 (1F, 1M) | Acoustic, auditory perceptual, visual imaging | Very low |
| Ikuma, Kunduk, Fink & McWhorter (2016) [45] | Solution validation with case studies | Clinical | 1 vocally healthy speaker, 1 with bilateral VF lesions and 1 with adductor spasmodic dysphonia | N = 3 (only the normophonic speaker had gender reported as female) | Visual imaging | Very low |
| Koike (1967) [37] | Cross-sectional | Clinical | Vocally healthy speakers and patients with laryngeal diagnoses such as cancer, unilateral laryngeal paralysis, laryngeal papilloma | Different numbers of subjects per experiment; Acoustic: controls n = 12 (10M, 2F), patients with laryngeal diseases n = 21 (14M, 7F); Aerodynamic: controls n = 12 (10M, 2F), pathologic larynges n = 11 (9M, 2F); Electromyographic: controls n = 2 (2M); Cineradiographic: controls n = 5 (5M), laryngeal lesions n = 7 (gender unspecified) | Acoustic, aerodynamic, auditory perceptual, physiological, visual imaging | Very low |
| Koster, Marx, Gemmar, Hess & Kunzel (1999) [30] | Cross-sectional | Laboratory | Vocally healthy speaker | N = 1, F aged 21 years | Acoustic, physiological, visual imaging | Very low |
| Kunduk (2004) [35] | Cross-sectional | Laboratory | Vocally healthy speakers in two age groups (younger and older) | N = 40 (20F per age group); Young: mean 26, range 18–35; Old: mean 76, range 68–82 years | Acoustic, visual imaging | Very low |
| Kunduk, Yan, McWhorter & Bless (2006) [46] | Cross-sectional | Laboratory | Vocally healthy speakers | N = 2 (2F), Mean = 50.5; SD = 27.5; Range = 23–78 | Visual imaging | Very low |
| Kunduk, Ikuma, Blouin & McWhorter (2017) [47] | Cross-sectional | Laboratory | Vocally healthy speaker | N = 1 (1F) | Visual imaging | Very low |
| Lebacq & DeJonckere (2019) [26] | Cross-sectional | Laboratory | Vocally healthy speaker | N = 1 (1M) | Acoustic, aerodynamic, physiological, visual imaging | Very low |
| Madill, Nguyen, McCabe, Ballard & Gregory (2019) [28] | Cross-sectional | Laboratory | Vocally healthy speakers | N = 30 (30F); age criteria 18–65 years | Acoustic, aerodynamic, physiological | Very low |

*(Continued)*

**Table 1.** (Continued)

| Study details | Study design | Setting | Characteristics of study population | Details of participants (age, gender, number) | Voice onset measurement categories explored | Certainty of evidence (GRADE) |
|---|---|---|---|---|---|---|
| Maryn & Poncelet (2021) [24] | Cross-sectional | Clinical | 6 patients no pathology, 3 with hypotrophy, 2 with vocal nodules, 2 with cyst, 1 oedema, 1 haemorrhage, 1 laryngitis, 1 granuloma, 1 bilateral VF paralysis, 1 unilateral VF paralysis | N = 20 (20F), Dutch-speaking, mean age: 57 years; age range: 23–87 years | Auditory perceptual | Very low |
| Mergell, Herzel, Wittenberg, Tigges & Eysholdt (1998) [48] | Solution validation with applied modelling and case study | Laboratory | Undefined human subject | N = 1 | Visual imaging | Very low |
| Naghibolhosseini, Zacharias, Zenas, Levesque & Deliyski (2023) [34] | Cross-sectional | Clinical | Vocally healthy speakers and speakers with adductor type spasmodic dysphonia (AdSD) | N = 10; 5 vocally healthy: 2M ranging in age from 35–49; 3F aged 35–67; 5 AdSD: 1M aged 76, 4F aged 60–76 | Visual imaging | Low |
| Orlikoff, Deliyski, Baken & Watson (2009) [11] | Cross-sectional | Laboratory | Vocally healthy speakers | N = 5 (3F, 2M) | Acoustic, physiological, visual imaging | Very low |
| Patel (2016) [49] | Cross-sectional | Laboratory | Vocally healthy speakers | N = 86; 43 children (n girls = 25, n boys = 18) (mean 8.2 years, range 5–11 years); 43 adults (n females = 25, n males = 18) (mean 23.6 years, range 21–45 years) | Visual imaging | Moderate |
| Patel, Walker & Dollinger (2017) [50] | Cross-sectional | Laboratory | Vocally healthy speakers | N = 71 (44F, 27M) | Visual imaging | Very low |
| Patel, Forrest, & Hedges (2017) [12] | Cross-sectional | Laboratory | Vocally healthy speakers | N = 56 (35F, 21M), Range = 20 to 42 years | Acoustic, visual imaging | Low |
| Peters, Boves & van Dielen (1986) [23] | Cross-sectional | Laboratory | Vocally healthy speakers | N = 4 (4M), mean age 20 years | Acoustic, auditory perceptual | Very low |
| Plant, Freed, & Plant (2004) [27] | Cross-sectional | Laboratory | Vocally healthy speakers | N = 5 (1F, 4M), mean age = 31, range = 26 to 47 | Acoustic, aerodynamic, physiological | Very low |
| Roark, Watson & Baken (2012) [51] | Validation study with cross-sectional data | Laboratory | Vocally healthy speakers | N = 112 (57F, 55M), Women (mean age: 28 years; range: 22–50 years); men (mean age: 29.1 years; range: 21–50 years) | Acoustic, physiological | Low |
| Roark, Watson, Baken, Brown & Thomas (2012) [52] | Cross-sectional | Laboratory | Vocally healthy speakers | N = 112 (57F, 55M), Women (mean age: 28 years; range: 22–50 years); men (mean age: 29.1 years; range: 21–50 years) | Acoustic, physiological | Low |
| Shiba & Chhetri (2016) [38] | Cross-sectional | Laboratory | Vocally healthy speakers | N = 27 | Acoustic, auditory perceptual, visual imaging | Very low |
| Simon & Maryn, (2022) [25] | Cohort | Laboratory | Undefined, however study indicates some of the participant voice samples from Maryn & Poncelet, 2020 study were included but does not define whether these patients had no pathology or a given voice disorder. Other samples were vocally healthy speakers | N = 11 auditory-perceptual raters, being university students of undefined age and gender. Training group = 6, control group = 5. Gender information only provided for two normophonic speakers (1F, 1M) | Auditory perceptual | Low |
| Tigges, Wittenberg, Mergell & Eysholdt (1999) [53] | Cross-sectional | Laboratory | Vocally healthy speaker | N = 1 (1M) | Visual imaging | Very low |

(*Continued*)

**Table 1.** (Continued)

| Study details | Study design | Setting | Characteristics of study population | Details of participants (age, gender, number) | Voice onset measurement categories explored | Certainty of evidence (GRADE) |
|---|---|---|---|---|---|---|
| Watson, Freeman & Dembowski (1991) [54] | Cross-sectional | Laboratory | Vocally healthy speakers | N = 30 (18F, 12M), Mean 30, SD 5, range 21–40 years | Acoustic | Very low |
| Watson, Baken, Roark, Reid, Ribeiro & Tsai (2013) [55] | Cross-sectional | Laboratory | Vocally healthy speakers | N = 13 (8F, 5M), Females: mean age: 31 years; range: 22–47 years, Males: mean age: 23 years; range: 22–26 years | Acoustic, physiological | Very low |
| Watson, Baken & Roark (2016) [56] | Cross-sectional | Laboratory | Vocally healthy speakers | N = 112 (57F, 55M), Women (mean age: 28 years; range: 22–50 years); men (mean age: 29.1 years; range: 21–50 years) | Acoustic, physiological | Low |
| Werner-Kukuk & von Leden (1970) [57] | Cross-sectional | Laboratory | Vocally healthy speakers | N = 4 (2F, 2M), age range 29–33 | Visual imaging | Very low |
| Wittenberg, Moser, Tigges & Eysholdt (1995) [58] | Solution validation with cross-sectional data | Laboratory | Vocally healthy speakers | N = 30, range 20–30 years | Visual imaging | Very low |
| Wittenberg, Mergell, Tigges & Eysholdt (1997) [59] | Solution validation with cross-sectional data | Clinical | 42 participants with hyperfunctional voice disorder, 10 with hypofunctional voice disorder | N = 52 subjects (16 male, 36 female) with hypo and hyperfunctional voice disorders, no information on age. Hyperfunctional = 36F, 6M; hypofunctional = 10M | Visual imaging | Very low |
| Wittenberg, Tigges, Mergell & Eysholdt (2000) [60] | Review paper with case study | Laboratory | Vocally healthy speaker | N = 1 (1M), aged 28 years | Visual imaging | Very low |

estimate), low (i.e. very likely that further research will have an important impact on our confidence in the estimate of effect and is likely to change the estimate), or very low (i.e. very uncertain about the estimate of effect). The GRADEpro app was used to facilitate this process and ensure that the abovementioned terms were informed by a consistent, systematic process [42–44].

## Results

### Process of identifying studies

The PRISMA flowchart in Fig 1 outlines the processes undertaken to collect and review the study records. Thirty-five records were identified as meeting the inclusion criteria for the review. Twenty-three studies involved visual imaging, 19 studies conducted acoustic analysis, 11 used physiological measures, seven studies involved auditory perceptual analysis, and four included aerodynamic analysis.

### Study design

Of the 35 studies included, 26 used a cross-sectional design, six were validation studies, two were review papers with single or multiple case examples and one was a cohort study. No study used a randomised controlled trial design.

### Study population characteristics

Table 1 presents an overview of each record included in the review, summarising study setting, participant characteristics, category of measurement and evidence certainty. It should be

**Table 2. Summary of collective study population characteristics.**

| Study population characteristic | Summary data |
| --- | --- |
| Sample size | Range (n): 1–112<br>Median (n): 13<br>>30 participants: 14 papers<br>≤5 participants: 13 papers<br>1 participant: 6 papers |
| Age | Unspecified: 15 papers<br>Paediatric participants: 1 paper (mean age: 8 years)<br>Adult participants: 19 papers (mean age: 20–76 years, reported in 16 studies) |
| Gender | Unspecified: 5 papers<br>Male or female only: 12 papers<br>Both genders: 18 papers |
| Voice disorder status | Normophonic participants: 28 studies<br>Speakers with voice disorders: 7 studies<br>Voice disorder diagnoses:<br> • Granuloma<br> • Haemorrhage<br> • Laryngeal cancer<br> • Laryngitis<br> • Spasmodic dysphonia<br> • Vocal hyper/hypofunction<br> • Vocal nodules<br> • Vocal fold oedema<br> • Vocal fold paralysis |
| Setting | Laboratory: 28 papers<br>Clinical: 7 papers |

noted that while some studies specified the setting in which their research took place, most settings could only be extrapolated from the study methodology. Studies which used data from only vocally healthy, normophonic speakers (i.e., non-patients) were classified as taking place in a laboratory setting. Studies which involved patients with some form of voice disorder diagnosis were classified as 'clinical'. However, only one study explicitly stated that patients were recruited directly from a voice clinical setting [34]. Table 2 offers a summary of study population characteristics across the collective paper set, including sample size, age, gender, vocal health status and setting.

## Voice onset types

A definition of voice onset was provided in 25 of the 35 studies (see S1 Checklist). Ten of these provided definitions of the specific voice onset measures used throughout the study (e.g., Vocal Attack Time), and 15 included the concept of voice onset being the period between the first adductory movement of the vocal folds and steady-state vibration. Twenty-one studies specifically examined different types of voice onset, namely being breathy (also referred to as 'aspirate'), normal (also referred to as 'comfortable', 'soft', 'easy' and 'modal') and hard (also referred to as 'glottal', 'pressed' and 'hard/glottal attack') voice onset types.

Whilst these are auditory perceptual classifications, not all studies compared or validated their instrumental measures with independently-rated auditory perceptual judgements, despite using voice onset type as a classification or identifier. Only three studies of the review set compared their instrumental measure to perceptual judgements. As auditory perceptual judgement of voice is considered the 'gold standard' of voice assessment [61], it is noteworthy that few studies used comparisons to auditory perceptual judgements to validate the measure being investigated.

Across the 35 studies, a wide range of voice onset measures were explored. Amongst these, some focused on a singular measure (e.g., laryngeal reaction time) whilst others examined one measure using several means of instrumentation, for example, Vocal Attack Time (VAT), which is measured using the vocal acoustic and glottographic signals. Other studies examined or compared several measures of voice onset. Overall, 39 different measures of voice onset were identified across the collective set. Our team mutually agreed that the best means of synthesising and presenting this heterogeneous data set was through grouping the studies according to their measurement approach. As such, the following categories of measurement were identified; acoustic, aerodynamic, auditory-perceptual analysis, physiological measures and visual imaging. In any case where a given study explored more than one category of measurement, it was included across all relevant categories. The collective findings across each of these measurement categories are outlined in the sections below.

## Voice onset measures

In total, there were 39 voice onset measures across the collective dataset. These are presented with their definitions in Table 3. These measures were developed and investigated using different methods of analysis, which are described in the following text.

## Categories of voice onset measurement

**a) Auditory perceptual analysis.** Auditory perceptual analysis involves a listener making an auditory judgement about the properties of a sound. In the case of voice onset studies, this judgement often relates to the type of onset produced. Seven of the 35 included studies involved auditory-perceptual analysis. All seven studies involved perceptual ratings of phonation onset type, ranging from soft to hard [23–25, 37], breathy to 'German' (a glottal plosive occurring in German classical singing) [36] and breathy to hard/ pressed [7, 38]. For four of the seven studies [7, 36–38], the auditory perceptual rating of samples was used only as a form of correlation to an instrumental measure of voice onset. This also served as confirmation that the participants had produced the onset types correctly before proceeding with other voice onset measurements; with 67% concordance between the attempted phonation type and rater in Shiba and Chhetri's study, 68% agreement reported in Cooke et al.'s paper, 80% of samples being correctly identified in Freeman et al.'s study and 100% agreement on attack types in Koike's study. Each of the studies explored different measures of voice onset, with three studies examining auditory perceptual judgements of voice onset as a voice onset measure in and of itself. Peters, Boves and Van Dielen, Maryn and Poncelet and Simon and Maryn's papers focused on auditory perceptual judgement of voice onset as a standalone voice onset measure, with Maryn and Poncelet and Simon and Maryn concluding that there was considerable variability both between and within raters regarding the perception of voice onset type. Meanwhile, Peters reported moderately high reliability of ratings ($r_{1.1} = 0.74$).

Automation of voice onset measurement was involved in four of the seven studies, and only in the processing and data generation stages for measures unrelated to auditory perceptual analysis. All seven studies performed some form of reliability analysis, which is presented in Table 4. Two studies conducted both inter- and intra-rater reliability [24, 25], with the remainder only exploring inter-rater reliability. Percentage agreement [7, 36–38], product-moment correlations [23], the intraclass correlation coefficient [24, 25] and Cohen's kappa [25] were the statistical measures used to calculate reliability. None of the seven papers explored sensitivity nor specificity of data obtained.

Collectively, the studies presented conflicting findings. Whilst Freeman et al., Peters et al. and Koike's papers suggested listeners could discriminate well between onset types, Cooke

**Table 3. Summary and definition of voice onset measures.**

| Name of voice onset measures | Definition | Studies |
|---|---|---|
| Air consumption | The amount of air that is used during the initial 200ms of phonation | [37] |
| Air leakage before vocalisation | The quantity of air that escapes prior to vocalisation across different voice onset types | [37] |
| Area transient duration | The transient duration of the glottal area waveform | [47] |
| Electroglottographic (EGG) signal analysis of voice onset | Analysis of the onset of voice through electroglottographic techniques | [30] |
| Exhalation to onset lag | The time lag from the beginning of exhalation to the onset of voicing | [28, 37] |
| First peak of the acoustic derivative waveform (ADW1) | The time between the first acoustic deviation and the first peak of the acoustic derivative | [28] |
| Frequency stabilisation time (FST)/ Settling time | The time taken to achieve a stable fundamental frequency across amplitude periodicity, time periodicity and glottal closure during voice onset. This can be represented graphically as a slope and/or area of stability beneath the slope to reflect the period between the onset of voicing and reaching a steady state of vocal fold vibration | [12, 32, 36, 38, 47] |
| Gesture duration of maximum distance | The duration of the vocal fold adductory gesture when moving from 80% to 20% of their maximum distance during adduction | [7] |
| Glottal area analysis | The analysis of glottal area configurations, glottal width and glottal area waveforms associated with voice onset | [30, 36, 49] |
| Glottal Attack Time (GAT) | The time between the first oscillation and first contact of the vocal folds at the onset of phonation | [34] |
| Interval between the first action potential and onset of sound | A measure of cricothyroid muscle activation as recorded through electromyography, being the interval between the first action potential of the muscle and the onset of voicing | [37] |
| Laryngeal airway resistance | A measure of the degree of glottalisation during voice production. This value tends to decrease as a speaker moves from normal to soft voicing and is larger in individuals with greater contraction of their vocal adductors during voicing | [27] |
| Laryngeal Reaction Time (LRT) | A means of measuring vocal motor control which involves measuring latency for a range of phonation onset types | [54] |
| Length transient duration | Transient duration of the vibrating glottal length | [47] |
| Maximum of the first derivative of the electroglottographic (EGG) signal (dEGG/$dt$) | A means of measuring peak closure speed of the vocal folds during voice onset | [27] |
| Maximum vocal fold velocity | The maximum speed at which the vocal folds move towards the midline during adduction at the onset of phonation | [7] |
| Number of cycles before steady state phonation | The number of detectable periodic oscillations preceding the steady state of phonation | [26] |
| Onset of the acoustic signal ($X_{1a}$) | The point in the waveform associated with the first periodic deviation from the zero baseline of the filtered acoustic signal relative to the onset of the acoustic file | [12] |
| Open quotient | A measure of the medial glottal width during voice onset, often calculated by dividing the fraction of the cycle during which the glottis is open over the duration of the entire cycle | [30, 57] |
| Oscillation amplitude patterns | The analysis of various patterns associated with the amplitude of vocal fold oscillation at voice onset, including the growth in vocal fold amplitude during the onset, attainment of full-length oscillation and the fluctuation in the envelope amplitude of pre-tracheal vibration | [23, 26, 31, 37, 45] |
| Oscillatory Onset Time (OOT) | Calculated in milliseconds from the entire glottal area waveform as the time interval between the first detectable oscillatory motion from the zero baseline and the first point at which amplitude periodicity was within two standard deviations of the subject's average steady-state value | [50] |
| Perceived abruptness of voice onset | The rating of voice onset type according to a nominal or ordinal scale based on a listener's auditory perceptual judgement | [7, 23–25, 36–38] |
| Phonation Onset Time (POT) | The duration of amplitude growth from 32.2% to 67.8% of initial to steady-state amplitude derived from the mid-membranous vocal fold trajectories for the left and right vocal folds | [38, 47, 50, 59] |
| Phonatory Posture Time (PPT) | Time from the final phonatory posture of the vocal folds to the onset of phonation | [38] |
| Phonation Threshold Pressure (PTP) | The threshold pressure necessary to sustain oscillation of the vocal folds | [27] |
| Pre-Phonatory Oscillations (PPOs) | The oscillations that occur in the vocal folds prior to phonation, observed from 0-100ms | [36] |
| Pressure- glottal area relationship | An element of the voice onset phenomenon, whereby the intraglottal pressure during the opening phase of the glottis exceeds that during the closing phase, which is the basic condition for an energy transfer from the lung pressure to the tissue of the vocal folds | [26] |

(*Continued*)

**Table 3.** (Continued)

| Name of voice onset measures | Definition | Studies |
|---|---|---|
| Ratios of maximum velocity to maximum distance | The ratios of maximum velocity to maximum distance between the vocal processes, which serves as an estimate of vocal fold stiffness during voice onset production | [7] |
| Speed quotient | The time of adduction or lateral excursion of the vocal folds during voice onset divided by the time of adduction or medial excursion | [57] |
| Time span from closed glottis to voice onset | The time span from the glottis moving from a closed position to the onset of voice. This can be collected through a cine-radiograph capture of laryngeal movements | [37] |
| Visual analysis of laryngeal configurations | An analysis of configurations of the larynx and vocal patterns when producing different forms of voice onset, as seen through high-speed imaging, transglottal photoglottography, airflow, electroglottography and ultrasonography | [26, 30, 33, 53, 57, 60] |
| Vocal Attack Time (VAT) | A measure of the duration of the onset phase of phonation. VAT presents a quantification of the time lag between the growth of sound pressure and vocal fold contact signals at the start of phonation. It represents the lag time of the peak in the cross-correlation function of the /a/ bandpass-filtered analytic sound pressure and electroglottographic signals during the period of vocal onset | [11, 28, 29, 51, 52, 55, 56] |
| Vocal fold adductory positioning | Analysis of patterns pertaining to the adduction of the vocal folds, including the degree of adduction of the vocal folds during different types of voice onset and maximum adduction of the vocal folds across different voice onset types | [36] |
| Vocal fold abductory positioning | Analysis of patterns pertaining to the abduction of the vocal folds, including maximum vocal fold abduction during vocal onsets | [36] |
| Vocal fold frequency pattern analysis | Analysis of the patterns and variation in frequency which occur during vocal fold oscillation segments at the onset of phonation | [26, 37, 47] |
| Vocal fold oscillation onset | Measures relating to the start of the first oscillation of the vocal folds. The observable initiation phase events involved in vocal fold oscillation onset include the start of the first oscillatory motion, time to initial medial contact of the vocal folds, oscillation reaching the full length of the vocal folds and oscillation reaching the steady-state amplitude | [7, 12, 45, 58] |
| Vocal Rise Time (VRT) | The time duration from the onset of sound to the point at which the envelope amplitude of the acoustic waves reaches the mean amplitude of the steady portion of phonation | [28, 37] |
| Voice Initiation Period (VIP) | A transient and chaotic vibratory period that precedes regular vocal fold vibration. This includes the time from the arytenoids making their first adductory movement for phonation onset to the achievement of regular vocal fold vibration | [35, 46, 50], |
| Voice Onset Coordination (VOC) | The time between the onset of airflow for exhalation to the onset of vocal fold adduction and voicing. This requires measurement of both EGG and airflow signals | [28] |

et al., Shiba and Chhetri, Maryn and Poncelet and Simon and Maryn's papers indicated that auditory perceptual judgement of voice onset type can be unreliable both within- and between-raters. Six of the studies reflect the lowest GRADE level of evidence with a rating of 'very low' evidence certainty and one with a rating of 'low' certainty of evidence. This low quality of research evidence combined with the variability in the findings of these studies calls into question the value of auditory perceptual judgements as the most accurate and reliable means of assessing voice onset in clinical contexts. A summary of data extracted from these auditory perceptual studies is provided in Table 5.

**b) Acoustic analysis.** Acoustic analysis involves examining the recorded voice signal characteristics related to speech and voice production. Amongst the studies included, 19 utilised acoustic analysis in their voice onset measurement procedures. A wide range of acoustic voice onset measures were explored, as summarised in Table 6, inclusive of Vocal Rise Time (VRT), the first peak of the acoustic derivative waveform (ADW1) and Laryngeal Reaction Time (LRT). Papers exploring most acoustically derived measures of voice onset did not typically provide numeric data for each onset type. Rather, these presented data ranging from descriptions of onset type differences, such as vibration and amplitude patterns, often in the absence of complete data reporting (for example, [30]), to small datasets regarding a new or uncommonly used voice onset measure. A common feature across all presented acoustic measures

**Table 4. Reliability, sensitivity and specificity data of included studies.**

| Study details | Reliability data reported | Sensitivity data reported | Specificity data reported |
|---|---|---|---|
| Baken & Watson (2019) [29] | No | No | No |
| Braunschweig, Flaschka, Schelhorn-Neise & Döllinger (2008) [31] | No | No | No |
| Choi, Oh & Choi (2015) [33] | Intra- and inter-rater reliability: conducted on 20% of samples. Cohen's kappa found inter-rater: 0.92, and intra-rater: 0.96 | No | No |
| Cohen, Cohen, Benyamini, Adi & Keshet (2019) [32] | No | No | No |
| Cooke, Ludlow, Hallett & Selbie (1997) [7] | Inter-rater reliability: 67.8% agreement of voice onset types produced. Labelling of the laryngeal points tested for consistency on all data. First field done 6 times on each subject and the coefficients of variation found. Median coefficient of variation ≤1.6% | No | No |
| Freeman, Woo, Saxman & Murry (2012) [36] | Inter-rater reliability: conducted on all data but lacking in detail. Paper states both raters attained 80% accuracy in perceptual detection of target onset type but inter-rater agreement values not provided | No | No |
| Ikuma, Kunduk, Fink & McWhorter (2016) [45] | No | No | No |
| Koike (1967) [37] | Inter-rater reliability: 5 trained judges correctly identified 100% of vocal attack types produced by 4 trained subjects (3 attack types each) | No | No |
| Koster, Marx, Gemmar, Hess & Kunzel (1999) [30] | No | No | No |
| Kunduk (2004) [35] | Inter and intra-rater reliability: assessed by re-analysing 10% of all tokens. Multivariate tests for each variable showed no significant differences between speech pathologists' measurement of VIP. Inter-rater reliability included assessing Wilk's lambda for different time periods as follows- T1: $F_{(3,7)} = 0.787$; T2: $F_{(3,9)} = 2.91$; T3: $F_{(3,11)} = 2.64$; T4: $F_{(3,11)} = 0.282$; Pre: $F_{(3,11)} = 2.42$; Post: $F_{(3,11)} = 0.347$ (all non-significant), while intra-rater reliability showed a significant correlation (0.975) | T4 and duration of TOTALT3T4 (timing characteristics of the VIP) were sensitive to the effects of aging of the laryngeal system, however the only measure reaching significance was the number of vibration cycles before full length vocal fold vibration achieved ($p = 0.001$) | No |
| Kunduk, Yan, McWhorter & Bless (2006) [46] | No | No | No |
| Kunduk, Ikuma, Blouin & McWhorter (2017) [47] | No | No | No |
| Lebacq & DeJonckere (2019) [26] | No | No | No |
| Madill, Nguyen, McCabe, Ballard & Gregory (2019) [28] | No | No | No |
| Maryn & Poncelet (2021) [24] | Inter and intra-rater reliability: Intra- conducted on 10% of samples, inter-on 100% of samples. 2-way random effects ICC model. Inter-rater: Low in general (R1 and R2: 0.145, R1 and R3: 0.245, R1 and R4: 0.460, R2 and R3: 0.301, R2 and R4: 0.197, R3 and R4: 0.410). Intra-rater: Variable- acceptable reliability for two, poor for other 2 raters (R1: 0.889, R2: 0.769, R3: 0.392, R4: 0.341) | No | No |
| Mergell, Herzel, Wittenberg, Tigges & Eysholdt (1998) [48] | No | No | No |

*(Continued)*

**Table 4.** (*Continued*)

| Study details | Reliability data reported | Sensitivity data reported | Specificity data reported |
|---|---|---|---|
| Naghibolhosseini, Zacharias, Zenas, Levesque & Deliyski (2023) [34] | Inter-rater reliability: Three raters conducted GAT and GOT measurement for several participants' data; the rater with the most reliable measurements completed the remainder of data measurement (Wald 99% confidence interval used to determine this). Measures by the 3 raters found to be reliable as the upper limit of the 99% confidence interval was <18 frames. This frame number criterion was considered acceptable given the range in F0 of the voices from 100-250Hz | No | No |
| Orlikoff, Deliyski, Baken & Watson (2009) [11] | No | No | No |
| Patel (2016) [49] | Intra-rater reliability: Random selection of 24 participants (12 child, 12 adult) for intra-rater reliability using Pearson product moment correlations, calculated for vibratory onset and offset for total GAW, left GAW and right GAW. Vibratory onset time for children for total, left and right GAW: 0.934, 0.958, 0.911; Vibratory onset time for adults for total, left and right GAW: 0.851, 0.844, 0.952. Results indicate high intra-rater reliability across all measures | No | No |
| Patel, Walker & Dollinger (2017) [50] | Intra-rater reliability: 20 participants (males = 10, females = 10) were randomly selected for intra-rater reliability for the OOT and OOToff measures. One rater performed the analysis for intra-rater reliability. Intra-rater Cronbach alpha (α) was calculated for OOT and OOToff. OOT showed good internal consistency: α = 0.880. OOToff showed excellent internal consistency: α = 0.931 | No | No |
| Patel, Forrest & Hedges (2017) [12] | Inter- and intra-rater reliability: 15 participants (10F, 5M) selected randomly to analyse inter- and intra-rater reliability. Pearson product-moment correlations and absolute difference calculated for the initial and subsequent measure for each variable. Pearson product-moment correlations for intra-rater reliability for X1a, X2a, X1g, X1.5g, X2g, X3g, X3.5g, and X4g exceeded 0.97 for all variables with an absolute difference for intra-rater comparisons of X1a, X2a, X1g, X1.5g, X2g, X3g, X3.5g, and X4g of 3, 3, 4, 1, 4, 7, 2, and 2 ms. Pearson product-moment correlations for inter-rater reliability X1a, X2a, X1g, X1.5g, X2g, X3g, X3.5g, and X4g ranged from 0.93 to 0.99 for all measures. Absolute differences for inter-rater comparisons of X1a, X2a, X1g, X1.5g, X2g, X3g, X3.5g and X4g were 3, 5, 6, 2, 7, 12, 5, and 4 ms | No | No |
| Peters, Boves & van Dielen (1986) [23] | Inter-rater reliability: Product-moment correlations of judges' ratings for all stimuli, with r1.1 = 0.74 | No | No |
| Plant, Freed & Plant (2004) [27] | No | No | No |
| Roark, Watson & Baken (2012) [51] | No | No | No |
| Roark, Watson, Baken, Brown & Thomas (2012) [52] | No | No | No |
| Shiba & Chhetri (2016) [38] | Inter-rater reliability: 2 raters rated all samples- inter-rater reliability calculated on accuracy of posture. No reliability on measurement. 82% accuracy for blinded phonatory rating; overall concordance of both raters with attempted target phonation 67%; Agreement per voice onset type: breathy = 74%, modal = 79%, pressed = 45% | No | No |

(*Continued*)

**Table 4.** (*Continued*)

| Study details | Reliability data reported | Sensitivity data reported | Specificity data reported |
|---|---|---|---|
| Simon & Maryn (2022) [25] | <u>Inter- and intra-rater reliability</u>: intra-rater repeated 20% of total voice samples each testing period, with intra-rater agreement determined through Cohen's K (nominal scale) and ICC (continuous scale). Intrarater values at pre-test: group 1 (trained) = 0.550 and group 2 = 0.592 (weak agreement). At post-test 1 values for group 1 = 0.378 (minimal agreement) and group 2 = 0.500 (weak agreement). At post-test 2, group 1 = 0.661 (moderate) and group 2 = 0.410 (weak). Intra-rater for nominal measures: pre-test group 1 ICC = 0.551 (moderate), group 2 = 0.617 (moderate); post-test 1 group 1 = 0.200 (poor), group 2 = 0.527 (moderate); post-test 2 group 1 = 0.372 (poor), group 2 = 0.318 (poor). Inter-rater for nominal scale: Group 1 at pre-test K = 0.173 (no agreement) and Group 2 K = 0.350 (minimal). At post-test 1, Group 1 mean = 0.232 and Group 2 mean = 0.302, (minimal). At post-test 2, Group 1 mean = 0.244 and Group 2 = 0.228, again (minimal). Continuous scale inter-rater reliability: pre-test Group 1 = ICC value of 0.195 (poor) and group 2 = 0.311 (poor). At post-test 1, Group 1 = 0.169, Group 2 = 0.378 (both poor). At post-test 2, Group 1 = 0.284, Group 2 = 0.201 ICC (poor reliability) | No | No |
| Tigges, Wittenberg, Mergell & Eysholdt (1999) [53] | No | No | No |
| Watson, Freeman & Dembowski (1991) [54] | <u>Inter-rater reliability</u>: 25% of trials were randomised and re-measured. Pearson r values indicate good agreement on a pair-by-pair basis, ranging from 0.819 for whispered /a/ to 0.998 for 'Oscar took Pete's cat' | No | No |
| Watson, Baken, Roark, Reid, Ribeiro & Tsai (2013) [55] | No | No | No |
| Watson, Baken & Roark (2016) [56] | No | No | No |
| Werner-Kukuk & von Leden (1970) [57] | No | No | No |
| Wittenberg, Moser, Tigges & Eysholdt (1995) [58] | No | No | No |
| Wittenberg, Mergell, Tigges & Eysholdt (1997) [59] | No | No | No |
| Wittenberg, Tigges, Mergell & Eysholdt (2000) [60] | No | No | No |

was the limited utility of applying these measures in clinical contexts, with many requiring specialised software or processes which would be expensive and/or impractical to complete during a clinical session.

No specifications of voice recording equipment were provided for model number nor brand for six studies and two studies provided no specification whatsoever of device used. An integrated microphone (I.e., a microphone integrated into a stroboscopy or similar system) was used in two studies, and a further three studies used an audiotape recorder. Only one of the devices was used across more than one study (RadioShack 33–3012 head-mounted microphone), however all three studies in which it was used involved a similar research team. Some form of automation was involved in the methodology of 14 of the included acoustic analysis studies, and usually this was across both the processing and data generation stages using

**Table 5. Voice onset and automation data for studies with auditory perceptual analysis.**

| Study details | Voice onset measure(s) used | Vocal tasks performed | Voice onset types investigated | Automation of processes (N/ Stage and algorithm/ software) |
|---|---|---|---|---|
| Cooke, Ludlow, Hallett & Selbie (1997) [7] | Gesture duration of maximum distance, maximum velocity, ratios of maximum velocity to maximum distance between vocal processes, vocal fold oscillation onset and perceived abruptness of voice onset | Produced /i/ with breathy, normal and hard onsets | Breathy, normal, hard | Processing and data: MATLAB script, 'Peak Performance Inc' |
| Freeman, Woo, Saxman & Murry (2012) [36] | Glottal area analysis, vocal fold adductory and abductory positioning, prephonatory oscillations (PPO), settling time and perceived abruptness of voice onset | Phonation of the vowel /i/ in modal register and in 5 gestures: breathy, normal, hard, staccato, and 'German' | Breathy, Normal, Hard, Staccato, 'German' | Processing and data: KayPentax Image Processing Software (KIPS) |
| Koike (1967) [37] | Air consumption, air leakage before vocalisation, exhalation to onset lag, interval between the first action potential and onset of sound, oscillation amplitude patterns, time span from closed glottis to voice onset, vocal fold frequency pattern analysis, Vocal Rise Time (VRT) and perceived abruptness of voice onset | Sustained phonation of /a/ using 3 types of vocal attack at optimal pitch level | Breathy, soft, hard | None |
| Maryn & Poncelet (2021) [24] | Perceived abruptness of voice onset | The word 'eerst' extracted from existing voice sample audio, specifically chosen to represent a range of voice onset hardness types | Onsets selected from existing dataset to represent the 'phonation onset continuum'., including breathy, soft and hard onsets | None |
| Peters, Boves & van Dielen (1986) [23] | Perceived abruptness of voice onset and oscillation amplitude patterns | /a/ then /u/ then /i/ in varying degrees of onset from hard to soft | 7 types of onset demonstrated visually through graphic representations of the amplitude envelopes, varying from soft to hard onset types. No written description provided; only graphic representations of each onset type in Fig 1 | Processing: Customised computer program |
| Shiba & Chhetri (2016) [38] | Phonation Onset time (POT), Phonatory Posture Time (PPT), Frequency Stabilisation Time (FST) and perceived abruptness of voice onset | /i/ sniff; model, breathy and pressed phonation of /i/ vowel | Breathy, modal, pressed | Processing: Phantom Camera Control Application software |
| Simon & Maryn (2022) [25] | Perceived abruptness of voice onset | The word 'eerst' extracted from existing voice sample audio and further anchor productions of 'eerst' specifically produced to reflect a range of voice onset types from breathy to hard | Continuum of onsets from breathy to hard: breathy-mild, breathy-moderate, breathy-severe, soft, hard-mild, hard-moderate, hard-severe | None |

software platforms and mathematical algorithms. Three of the included papers used auditory perceptual analysis as a means of validating the instrumental measures used [36–38].

Only seven studies reported reliability assessment of acoustic analysis, of which two explored both inter- and intra-rater reliability [12, 35] and five explored only inter-rater reliability [23, 36–38, 54]. The following statistical methods were used to determine reliability across the studies: Pearson product-moment correlation [12, 23, 51, 54–56], percentage agreement [36, 37] and multivariate tests [35]. One of the studies that used acoustic measurements for voice onset reported sensitivity analysis [35], with none conducting an analysis on specificity.

In summary, the included acoustic analysis studies reflected low evidence certainty, with outcomes from the GRADE Certainty Assessment yielding a 'very low' rating for 14 studies,

**Table 6. Voice onset and automation data for studies with acoustic analysis.**

| Study details | Voice onset measure(s) used | Vocal tasks performed | Voice onset types investigated | Automation of processes (N/ Stage and algorithm/ software) |
|---|---|---|---|---|
| Baken & Watson (2019) [29] | Vocal Attack Time (VAT) | Not specified in this paper- data applied here was drawn from several previous studies conducted by the research team, each of which involved different phonatory tasks | Normal 'soft' onset, 'hard attack' | None |
| Braunschweig, Flaschka, Schelhorn-Neise & Döllinger (2008) [31] | Analysis of oscillation amplitude patterns | Phonation of /i/ for 3–4 seconds while maintaining a constant pressure level | No specific voice onset types examined- this study was examining the difference in voice outcomes between vocally healthy subjects and those with functional dysphonia | Processing and data: Mathematical formulas, algorithms and image analysis processes |
| Cohen, Cohen, Benyamini, Adi & Keshet (2019) [32] | Settling time, including slope from phonation onset to settling time and area under the fundamental frequency curve from phonation onset to settling time | Production of /i/ from silence | No specific voice onset types- subjects asked to produce sustained vowel /i/ from complete silence | Data: Praat script |
| Freeman, Woo, Saxman & Murry (2012) [36] | Glottal area analysis, vocal fold adductory and abductory positioning, prephonatory oscillations (PPO), settling time and perceived abruptness of voice onset | Phonation of the vowel /i/ in modal register and in 5 gestures: breathy, normal, hard, staccato, and 'German' | Breathy, Normal, Hard, Staccato and 'German' | Processing and data: KayPentax Image Processing Software (KIPS) |
| Koike (1967) [37] | Air consumption, air leakage before vocalisation, exhalation to onset lag, interval between the first action potential and onset of sound, oscillation amplitude patterns, time span from closed glottis to voice onset, vocal fold frequency pattern analysis, Vocal Rise Time (VRT) and perceived abruptness of voice onset | Sustained phonation of /a/ using 3 types of vocal attack at optimal pitch level | Breathy, soft and hard | None |
| Koster, Marx, Gemmar, Hess & Kunzel (1999) [30] | EGG signal analysis, glottal area analysis, open quotient and visual analysis of laryngeal configurations | One-off productions of /ihi/ and /ɛhɛ/ | Breathy and physiologic | Processing and data: Multi-dimensional Voice Analysis System (MVAS) |
| Kunduk (2004) [35] | Voice Initiation Period (VIP) | Vowel /i/, normal pitch and loudness at a comfortable speed | Nothing specified beyond 'comfortable, normal pitch and loudness level' | Processing and data: KayElemetrics HSDI, KIPS and Yan methods |
| Lebacq & DeJonckere (2019) [26] | Number of cycles before steady state phonation, oscillation amplitude patterns, pressure-glottal area relationship, visual analysis of laryngeal configurations and vocal fold frequency pattern analysis | A sustained vowel or a syllable beginning with a vowel | Breathy, soft and hard | None |
| Madill, Nguyen, McCabe, Ballard & Gregory (2019) [28] | The first peak of the acoustic derivative waveform (ADW1), VAT, VRT and VOC | Production of /a, i, ou/ at habitual pitch and loudness | Onset not specified; only vowels /a, i, ou/ at comfortable pitch and loudness | Data: LabChart, MATLAB and Praat script |
| Orlikoff, Deliyski, Baken & Watson (2009) [11] | Vocal Attack Time (VAT) | Subjects provided several samples of phonation which, in their estimation, involved breathy, soft/ comfortable and hard vocal onsets | Breathy, soft/comfortable and hard | Processing and data: custom MATLAB script |
| Patel, Forrest & Hedges (2017) [12] | Frequency stabilisation time, onset of the acoustic signal (X1a) and vocal fold oscillation onset | At least 3 consecutive /hi/ productions at self-selected pitch and loudness | Aimed to obtain the true physiological onset of the vocal folds through production of consecutive repetitions of /hi/. No specific onset types stipulated | Processing and data: Glottal Analysis Tools, MATLAB and Praat script |

*(Continued)*

**Table 6.** (*Continued*)

| Study details | Voice onset measure(s) used | Vocal tasks performed | Voice onset types investigated | Automation of processes (N/ Stage and algorithm/ software) |
|---|---|---|---|---|
| Peters, Boves & van Dielen (1986) [23] | Perceived abruptness of voice onset and oscillation amplitude patterns | /a/ then /u/ then /i/ in varying degrees of onset from hard to soft | 7 types of onset demonstrated visually through graphic representations of the amplitude envelopes, varying from soft to hard onset types. No written description provided; only graphic representations of each onset type in Fig 1 | Processing: Customised computer program |
| Plant, Freed & Plant (2004) [27] | Laryngeal airway resistance, Maximum of the first derivative of the EGG signal (dEGG/dt) and Phonation Threshold Pressure (PTP) | Phonation of /i/. Specific tasks given were to 1) say 3 tokens with a breathy onset; 2) say 3 tokens at low pitch with the 1st token at low intensity and the next two at increasing intensity; 3) say 3 tokens at mid-pitch, increasing in intensity; 4) say 3 tokens at high pitch, increasing in intensity; and 5) say 3 tokens with a glide from low to high pitch | Only one production task specified the target onset type as 'breathy' | None. |
| Roark, Watson & Baken (2012) [51] | Vocal Attack Time (VAT) | 'Always', 'hallways', sustained phonation of vowel /a/ | Only specified 'comfortable pitch, loudness and rate | None. |
| Roark, Watson, Baken, Brown & Thomas (2012) [52] | Vocal Attack Time (VAT) | 'Always', 'hallways', sustained phonation of vowel /a/ | Only specified 'comfortable pitch, loudness and rate' | Processing and data: Baken-Orlikoff method |
| Shiba & Chhetri (2016) [38] | Phonation Onset time (POT), Phonatory Posture Time (PPT), Frequency Stabilisation Time (FST) and perceived abruptness of voice onset | /i/ sniff; model, breathy and pressed phonation of /i/ vowel | Breathy, modal, pressed | Processing: Phantom Camera Control Application software |
| Watson, Freeman & Dembowski (1991) [54] | Laryngeal Reaction Time (LRT) | Cough, whispered /a/, voiced /a/, 'Oscar' and 'Oscar took Pete's cat' | Aspirate, normal and hard | Processing and data: Digitised signal was software rectified and analysed |
| Watson, Baken, Roark, Reid, Ribeiro & Tsai (2013) [55] | Vocal Attack Time (VAT) | Sustained phonations of /a/, /i/, and /u/ produced with low, mid, and high F0 | Only specified to produce each vowel at comfortable pitch and loudness | Processing and data: Baken-Orlikoff method, E-system software speech synthesiser and MATLAB customised algorithms |
| Watson, Baken & Roark (2016) [56] | Vocal Attack Time (VAT) | Sustained phonation of vowel /a/ and production of the words "always" and "hallways" | Only specified to produce each vowel at comfortable pitch and loudness | Processing and data: Baken-Orlikoff method |

and a rating of 'low' for the remaining five. While a large proportion of the reviewed studies involved acoustic analysis measures, there is evidently a vast range of acoustic analysis measures being used which prevents an in-depth understanding of any given measure. The acoustic analysis findings overall cannot be interpreted with high levels of confidence, nor are they of sufficient quality to inform the selection of the most reliable, sensitive, and specific acoustic voice onset measures for clinical practice.

**c) Aerodynamic analysis.** Aerodynamic analysis refers to the measurement of phonatory airflow, volume, pressure and combined measures, such as efficiency and resistance. Four papers reported airflow measurement information informing some aspect of voice onset. The specific airflow measures explored across these studies included air consumption during the initial 200 milliseconds of different attack types (soft, breathy and hard) [37], Phonation

Threshold Pressure (PTP) [27], Voice Onset Coordination (VOC) [28] and vocal onset according to transglottal airflow and intraglottal pressure [26].

Koike and LeBacq and DeJonckere's papers similarly focused their studies upon exploring the characteristics of different voice onsets. Koike identified that soft and hard onsets were diametrically opposed across a range of measures, while the breathy onset showed little relation to either, having a 'distinct character' that differed completely from soft and hard onset types. LeBacq and DeJonckere namely used their airflow data as part of an intraglottal pressure calculation, while Madill et al.'s study correlated existing voice onset measures, including VOC, with the measure ADW1, concluding that it can be predicted from VOC. In Plant's exploration of phonation threshold pressure, it was found that for most subjects, increasing airway resistance coincided with increasing threshold pressure.

Devices used for airflow measures were largely consistent, with three of the four studies using a Rothenberg mask or equivalent, and the other paper using a pneumatochograph [37]. Most of the four papers didn't involve any automated processes, apart from Madill's study, which involved some automation only in the data generation phase. Only Koike's study involved some form of reliability assessment, being inter-rater reliability established through percentage agreement. None of the studies performed an analysis of sensitivity nor specificity. None of the included papers used auditory perceptual ratings to validate the instrumental measures of voice onset used.

Overall, the aerodynamic data presented across these four studies did not contribute significantly to an understanding of the most effective means of assessing voice onset through airflow. Other than Koike and Madill, there is a lack of transparency when it comes to presentation of the aerodynamic voice onset data. These findings should be considered as offering indefinite conclusions pertaining to the value of aerodynamic voice onset measurement, particularly as all four studies were graded as having the lowest certainty of evidence, being 'very low' evidence certainty according to the GRADE rating system. A summary of these aerodynamic analysis studies is provided in Table 7.

**d) Physiological measures.** A range of other instrumental measures that monitor physiological muscle movement have been used to measure voice onset. For the purposes of this review, this specifically relates to electroglottography (EGG) and electromyography (EMG). EGG is a non-invasive technology used to measure the varying degrees of vocal fold contact during voice production, while EMG is a measure of muscular response or activation. Eleven studies explored physiological measures of voice onset. The specific types of voice onset measures examined in these studies included VAT [11, 28, 29, 51, 52, 55, 56], maximum of the first derivative of the EGG signal [27] and the interval between the first action potential (as detected by EMG) and the onset of sound [37].

The three studies of low evidence were largely conducted by the same research group [51, 52, 56], and all explored VAT as a measure of voice onset. However, the research questions posed in each of these studies differed, ranging from determining the fidelity of VAT as a voice onset measure to establishing normative VAT values. Pearson's correlation coefficient was found to be a suitable fidelity metric (median correlation coefficient of 0.975 for 1033 VAT measures) [51], with the mean VAT among healthy young adults reported as 1.98ms. Aspirated voice onsets (e.g., the production of 'hallways') lead to a greater mean VAT than unaspirated voice onset tasks (e.g., the production of 'always') [56]. All remaining studies were of 'very low' evidence certainty; the majority of which also explored VAT.

Devices used across the physiological studies were varied, with three studies providing no specification of equipment. The remaining eight studies included one electromyograph and the remainder a combination of electroglottographs of different brands and models, with only the Glottal Enterprises EG2 and the KayPENTAX Fourcin Laryngograph model 6091

**Table 7.** Voice onset and automation data for studies with aerodynamic analysis.

| Study details | Voice onset measure(s) used | Vocal tasks performed | Voice onset types investigated | Automation of processes (N/ Stage and algorithm/ software) |
|---|---|---|---|---|
| Koike (1967) [37] | Air consumption, air leakage before vocalisation, exhalation to onset lag, interval between the first action potential and onset of sound, oscillation amplitude patterns, time span from closed glottis to voice onset, vocal fold frequency pattern analysis, Vocal Rise Time (VRT) and perceived abruptness of voice onset | Sustained phonation of /a/ using 3 types of vocal attack at optimal pitch level | Breathy/ aspirate, soft/ simultaneous and hard/ glottal | None |
| Lebacq & DeJonckere, (2019) [26] | Number of cycles before steady state phonation, oscillation amplitude patterns, pressure-glottal area relationship, visual analysis of laryngeal configurations and vocal fold frequency pattern analysis | A sustained vowel or a syllable beginning with a vowel | Breathy, soft and hard onsets | None |
| Madill, Nguyen, McCabe, Ballard & Gregory (2019) [28] | The first peak of the acoustic derivative waveform (ADW1), VAT, VRT and VOC | Production of /a, i, ou/ at habitual pitch and loudness | Onset not specified; only vowels /a, i, ou/ at comfortable pitch and loudness | Data: LabChart, MATLAB and Praat script |
| Plant, Freed & Plant (2004) [27] | Laryngeal airway resistance, Maximum of the first derivative of the EGG signal (dEGG/dt) and Phonation Threshold Pressure (PTP) | Phonation of /i/. Specific tasks given were to 1) say 3 tokens with a breathy onset; 2) say 3 tokens at low pitch with the 1st token at low intensity and the next two at increasing intensity; 3) say 3 tokens at mid-pitch, increasing in intensity; 4) say 3 tokens at high pitch, increasing in intensity; and 5) say 3 tokens with a glide from low to high pitch | Only one production task specified the target onset type as 'breathy' | None |

occurring in more than one study (each used in two studies). Six of the included studies involved automation as part of their study methodology for physiological measures, with five of these employing automated processes or algorithms across both the data processing and generation stages and one only using automation for data generation.

Reliability analysis was only performed in one study; Koike, 1967, which conducted inter-rater reliability as determined through percentage agreement (see Table 4). Neither sensitivity nor specificity analysis was conducted in any of the papers within this category. Only one of the papers in this set included auditory perceptual analysis to validate the instrumental measures of voice onset used.

Collectively, the studies of the highest GRADE level of evidence examining physiological measures of voice onset use VAT. Despite the greater breadth of research upon VAT than most other voice onset measures, there is a requirement to collect both electroglottographic and acoustic data to attain the VAT value. This, combined with the limited availability of the MATLAB-based program to calculate the measure, the heterogeneity amongst research questions posed in these studies, and the highest evidence rating according to the GRADE rating system as 'low' calls into question its clinical utility. A summary of the studies involving physiological analysis is provided in Table 8.

**e) Visual imaging.** Visual imaging relates to any study whereby a measure of voice onset was based upon still or motion pictures of the larynx. Amongst the 35 included studies, 23 involved visual imaging in their measurement of voice onset. These studies investigated a range of measures related to voice onset, including Phonation Onset Time (POT) [38, 48, 50, 59], measures of velocity, angle, distance and time associated with voice onset [7], Voice Initiation Period (VIP) [35, 46, 50] and Glottal Attack Time (GAT) [34].

**Table 8. Voice onset and automation data for studies with physiological analysis.**

| Study details | Voice onset measure(s) used | Vocal tasks performed | Voice onset types investigated | Automation of processes (N/ Stage and algorithm/ software) |
|---|---|---|---|---|
| Baken & Watson (2019) [29] | Vocal Attack Time (VAT) | Not specified in this paper- data applied here was drawn from several previous studies conducted by the research team, each of which involved different phonatory tasks | Normal 'soft' onset, 'hard attack' | None |
| Koike (1967) [37] | Air consumption, air leakage before vocalisation, exhalation to onset lag, interval between the first action potential and onset of sound, oscillation amplitude patterns, time span from closed glottis to voice onset, vocal fold frequency pattern analysis, Vocal Rise Time (VRT) and perceived abruptness of voice onset | Sustained phonation of /a/ using 3 types of vocal attack at optimal pitch level | Breathy, soft and hard | None |
| Koster, Marx, Gemmar, Hess & Kunzel, (1999) [30] | EGG signal analysis, glottal area analysis, open quotient and visual analysis of laryngeal configurations | One-off productions of /ihi/ and /ƐhƐ/ | Breathy and physiologic | Processing and data: Multi-dimensional Voice Analysis System (MVAS) |
| Lebacq & DeJonckere (2019) [26] | Number of cycles before steady state phonation, oscillation amplitude patterns, pressure-glottal area relationship, visual analysis of laryngeal configurations and vocal fold frequency pattern analysis | A sustained vowel or a syllable beginning with a vowel | Breathy, soft and hard onsets | None |
| Madill, Nguyen, McCabe, Ballard & Gregory (2019) [28] | The first peak of the acoustic derivative waveform (ADW1), VAT, VRT and VOC | Production of /a, i, ou/ at habitual pitch and loudness | Onset not specified; only vowels /a, i, ou/ at comfortable pitch and loudness | Data: LabChart, MATLAB and Praat script |
| Orlikoff, Deliyski, Baken & Watson (2009) [11] | Vocal Attack Time (VAT) | Subjects provided several samples of phonation which, in their estimation, involved breathy, soft/ comfortable and hard vocal onsets | Breathy, soft/ comfortable and hard | Processing and data: custom MATLAB script |
| Plant, Freed & Plant (2004) [27] | Laryngeal airway resistance, Maximum of the first derivative of the EGG signal (dEGG/dt) and Phonation Threshold Pressure (PTP) | Phonation of /i/. Specific tasks given were to 1) say 3 tokens with a breathy onset; 2) say 3 tokens at low pitch with the 1st token at low intensity and the next two at increasing intensity; 3) say 3 tokens at mid-pitch, increasing in intensity; 4) say 3 tokens at high pitch, increasing in intensity; and 5) say 3 tokens with a glide from low to high pitch | Only one production task specified the target onset type as 'breathy' | None |
| Roark, Watson & Baken (2012) [51] | Vocal Attack Time (VAT) | 'Always', 'hallways', sustained phonation of vowel /a/ | Only specified 'comfortable pitch, loudness and rate | None |
| Roark, Watson, Baken, Brown & Thomas (2012) [52] | Vocal Attack Time (VAT) | 'Always', 'hallways', sustained phonation of vowel /a/ | Only specified 'comfortable pitch, loudness and rate' | Processing and data: Baken-Orlikoff method |
| Watson, Baken, Roark, Reid, Ribeiro & Tsai (2013) [55] | Vocal Attack Time (VAT) | Sustained phonations of /a/, /i/, and /u/ produced with low, mid, and high F0 | Only specified to produce each vowel at comfortable pitch and loudness | Processing and data: Baken-Orlikoff method, E-system software speech synthesiser and MATLAB customised algorithms |
| Watson, Baken & Roark (2016) [56] | Vocal Attack Time (VAT) | Sustained phonation of vowel /a/ and utteration of the words "always" and "hallways" | Only specified to produce each vowel at comfortable pitch and loudness | Processing and data: Baken-Orlikoff method |

Twenty of the 23 studies in this category involved high speed visual imaging, with kymography used in five studies [11, 36, 38, 53, 60], rigid laryngoscopy in one [7] and one employing cine-radiographic techniques, i.e., the recording of laryngeal movements on x-ray film [37]. Devices used across the visual imaging studies were varied, with the most common device used being the KayPENTAX colour high speed video system and component model 9710, used in five of the 23 studies. Six studies did not specify the device used, and of the remaining studies, 11 used some form of high-speed camera system and the remaining study performed cineradiography. Nineteen studies utilised a software program or mathematical algorithm to automate the processing and/or analysis of data pertaining to vocal fold vibration and glottal characteristics (Table 9).

Ten studies used reliability assessment in their measurement protocols, involving three which explored both inter- and intra-rater reliability [12, 33, 35], five inter-rater [7, 34, 36–38] and two intra-rater reliability assessments [49, 50]. The statistical methods used for reliability assessment included Pearson product moment correlations [12, 49], Cohen's kappa [33], Cronbach alpha [50], Pearson's correlation coefficient, general linear model and repeated measures analysis [35], the Wald 99% confidence interval [34] and percentage agreement [7, 36–38]. Most studies did not report any sensitivity assessment, except for one paper [35]. No studies conducted specificity analysis. Three of the papers which involved visual imaging included auditory perceptual analysis to validate the instrumental measures of voice onset used.

Regarding the GRADE Certainty Assessment, one study was rated as 'moderate', two as 'low', and 20 as 'very low' certainty of evidence. The findings of this section prove that the use of equipment (namely laryngoscopy) can introduce further variance in voice onset measures used, with an extensive range of voice onset measures despite the similarities across the visual imaging hardware used.

## Automated voice onset measures

In examining the 35 studies, an interesting theme which arose was the increasing use of task automation to obtain voice onset measures in recent years. For the purposes of this review, 'automation' refers to any process throughout a study's methodology which uses a form of computerised software or algorithm to eliminate the manual need to prepare or process data. Only nine studies [24–27, 29, 37, 51, 57, 60] were found to involve no automated processes. These studies generally involved a research question focused upon auditory perceptual judgements, reliability or fidelity checking, or presented a descriptive review of a specific voice onset measure based on previous literature, and as such did not involve the analysis of large sets of objective voice onset measurement data. There were four studies which only involved automation in the pre-processing phase [23, 33, 34, 38], with most using an automated process for both pre-processing and/or voice onset data output. Three studies used automation for data output alone [28, 32, 49]. According to measurement category, those studies which fell within the visual imaging and acoustic categories mainly used automation for processing and data. Across the remaining categories of physiological, aerodynamic and auditory perceptual studies, the automated phases of data analysis tended to vary more greatly.

Across the 26 studies which used automated algorithms, 12 used solely proprietary software or programs to perform automated functions upon their datasets, nine used only customised algorithms or programs, three used a combination of either proprietary and custom software, or used proprietary software with customised algorithms or applications specific to the research project and two were unspecified/unclear. There were several proprietary tools used across multiple studies, with the most common being MATLAB, in seven papers. While

**Table 9. Voice onset and automation data for studies with visual imaging analysis.**

| Study details | Voice onset measure(s) used | Vocal tasks performed | Voice onset types investigated | Automation of processes (N/ Stage and algorithm/ software) |
|---|---|---|---|---|
| Braunschweig, Flaschka, Schelhorn-Neise & Döllinger (2008) [31] | Analysis of oscillation amplitude patterns | Phonation of /i/ for 3–4 seconds while maintaining a constant pressure level | No specific voice onset types examined- this study was examining the difference in voice outcomes between vocally healthy subjects and those with functional dysphonia | Processing and data: Mathematical formulas, algorithms and image analysis processes |
| Choi, Oh & Choi (2015) [33] | Visual analysis of laryngeal configurations | Sustained vowel /i/ | Types I, II and III | Processing: Photron FastCam Viewer |
| Cooke, Ludlow, Hallett & Selbie (1997) [7] | Gesture duration of maximum distance, maximum velocity, ratios of maximum velocity to maximum distance between vocal processes, vocal fold oscillation onset and perceived abruptness of voice onset | Produced /i/ with breathy, normal and hard onsets | Breathy, normal, hard | Processing and data: MATLAB script, 'Peak Performance Inc' |
| Freeman, Woo, Saxman & Murry (2012) [36] | Glottal area analysis, vocal fold adductory and abductory positioning, prephonatory oscillations (PPO), settling time and perceived abruptness of voice onset | Phonation of the vowel /i/ in modal register and in 5 gestures: breathy, normal, hard, staccato, and 'German' | Breathy, Normal, Hard, Staccato and 'German' | Processing and data: KayPentax Image Processing Software (KIPS) |
| Ikuma, Kunduk, Fink & McWhorter (2016) [45] | Vocal fold oscillation onset, oscillation amplitude patterns | Sustained phonation of a speaker with a bilateral lesion, another of a speaker with adductor type spasmodic dysphonia (ADSD), at normal pitch and loudness and with imitated breathiness. Nil specification beyond this provided | None specifically under investigation, however a range of onset type exemplars were applied to the algorithm to gauge its utility in detecting initiation phase events in different onsets: 'normal pitch and loudness', 'imitated breathy voice', bilateral lesions and ADSD | Processing and data: Transient detection algorithm |
| Koike (1967) [37] | Air consumption, air leakage before vocalisation, exhalation to onset lag, interval between the first action potential and onset of sound, oscillation amplitude patterns, time span from closed glottis to voice onset, vocal fold frequency pattern analysis, Vocal Rise Time (VRT) and perceived abruptness of voice onset | Sustained phonation of /a/ using 3 types of vocal attack at optimal pitch level | Breathy, soft and hard | None |
| Koster, Marx, Gemmar, Hess & Kunzel (1999) [30] | EGG signal analysis, glottal area analysis, open quotient and visual analysis of laryngeal configurations | One-off productions of /ihi/ and /ƐhƐ/ | Breathy and physiologic | Processing and data: Multi-dimensional Voice Analysis System (MVAS) |
| Kunduk (2004) [35] | Voice Initiation Period (VIP) | Vowel /i/, normal pitch and loudness at a comfortable speed | Nothing specified beyond 'comfortable, normal pitch and loudness level' | Processing and data: KayElemetrics HSDI, KIPS and Yan methods |
| Kunduk, Yan, McWhorter & Bless (2006) [46] | Voice Initiation Period (VIP) | Short, successive productions of vowel /i/ at normal pitch, loudness and comfortable speed | Nothing specified beyond productions at comfortable, normal pitch and loudness | Processing and data: Transient detection algorithm |

*(Continued)*

**Table 9.** (*Continued*)

| Study details | Voice onset measure(s) used | Vocal tasks performed | Voice onset types investigated | Automation of processes (N/ Stage and algorithm/ software) |
|---|---|---|---|---|
| Kunduk, Ikuma, Blouin & McWhorter (2017) [47] | Area transient duration, length transient duration, settling time and vocal fold frequency pattern analysis | Production of voices with varying pitch and loudness, including normal pitch normal loudness (NPNL), normal pitch loud (NPL), high pitch and comfortable loudness (HPNL) and high pitch and loud (HPL). The subject also imitated breathy and pressed phonation using comfortable volume and pitch | Produced voices of 'varying pitch and loudness', with two onset types specified: Comfortable pitch and loudness, normal pitch and loudness, high pitch and comfortable loudness, high pitch and loud, breathy and pressed voice with comfortable volume and pitch | Processing and data: Algorithm |
| Lebacq & DeJonckere (2019) [26] | Number of cycles before steady state phonation, oscillation amplitude patterns, pressure-glottal area relationship, visual analysis of laryngeal configurations and vocal fold frequency pattern analysis | A sustained vowel or a syllable beginning with a vowel | Breathy, soft and hard | None |
| Mergell, Herzel, Wittenberg, Tigges & Eysholdt (1998) [48] | Phonation Onset Time (POT) | Two-mass computer simulated model considered adequate to study the basic features of phonation onset. Later application to high-speed electroglottography | Onset types not specified | Processing and data; Wittenberg's semi-automatic motion analysis software |
| Naghibolhosseini, Zacharias, Zenas, Levesque & Deliyski (2023) [34] | Glottal Attack Time (GAT) | Subjects only instructed to read the 6 CAPE-V phrases and first 6 sentences of the Rainbow Passage | Onset types not specified | Processing: Phantom Camera Control Application software |
| Orlikoff, Deliyski, Baken & Watson (2009) [11] | Vocal Attack Time (VAT) | Subjects provided several samples of phonation which, in their estimation, involved breathy, soft/ comfortable and hard vocal onsets | Breathy, soft/ comfortable and hard | Processing and data: custom MATLAB script |
| Patel (2016) [49] | Glottal area analysis | /hi/ productions at self-selected conversational pitch and loudness | Aimed to obtain the 'true vibratory onset' of the vocal folds through production of consecutive repetitions of /hi/ | Data: Glottal Analysis Tool |
| Patel, Walker & Dollinger (2017) [50] | Oscillatory Onset Time (OOT), Phonation Onset Time (POT) and Voice Initiation Period (VIP) | Consecutive productions of /hi/ | Aimed to obtain the true physiological onset of the vocal folds through production of consecutive repetitions of /hi/ | Processing and data: Glottal analysis tool, MATLAB |
| Patel, Forrest & Hedges (2017) [12] | Frequency stabilisation time, onset of the acoustic signal (X1a) and vocal fold oscillation onset | At least 3 consecutive /hi/ productions at self-selected pitch and loudness | Aimed to obtain the true physiological onset of the vocal folds through production of consecutive repetitions of /hi/ | Processing and data: Glottal Analysis Tools, MATLAB and Praat script |
| Shiba & Chhetri (2016) [38] | Phonation Onset time (POT), Phonatory Posture Time (PPT), Frequency Stabilisation Time (FST) and perceived abruptness of voice onset | /i/ sniff; model, breathy and pressed phonation of /i/ vowel | Breathy, modal and pressed | Processing: Phantom Camera Control Application software |
| Tigges, Wittenberg, Mergell & Eysholdt (1999) [53] | Visual analysis of laryngeal configurations | Vowel /i/ in normal, hard, and breathy onsets | Breathy, normal and hard | Processing and data: Digital line scanning/kymograph algorithm |
| Werner-Kukuk & von Leden (1970) [57] | Open quotient, speed quotient and visual analysis of laryngeal configurations | Soft, breathy and hard attacks of /i/ vowel at comfortable pitch and loudness | Breathy, soft and hard | None |
| Wittenberg, Moser, Tigges & Eysholdt (1995) [58] | Vocal fold oscillation onset measures | Vowel /i/ three times using normal, soft and hard vocal onsets | Soft, normal and hard | Processing and data: Image processing algorithm |

(*Continued*)

**Table 9.** (*Continued*)

| Study details | Voice onset measure(s) used | Vocal tasks performed | Voice onset types investigated | Automation of processes (N/ Stage and algorithm/ software) |
|---|---|---|---|---|
| Wittenberg, Mergell, Tigges & Eysholdt (1997) [59] | Phonation Onset Time (POT) | Production of epsilon / ɛ/ at phonatory ease | Soft, normal and hard | Processing and data: In house software/ semi-automated motion analysis algorithm |
| Wittenberg, Tigges, Mergell & Eysholdt (2000) [60] | Visual analysis of laryngeal configurations | Vowel /i/ using normal, hard, and breathy onsets | Breathy, normal and hard | This paper is descriptive in nature. It described how automation could be applied in the processing and data phases, but did not implement these automated processes itself |

certain algorithms and filters were also named and described across studies, a close examination of these is beyond the scope of this paper.

## Research quality

The process of data extraction included extracting data pertaining to the conduction of reliability, sensitivity or specificity analysis in any of the 35 studies. It was found that fourteen of the 35 studies conducted some form of reliability analysis while one conducted some form of sensitivity or specificity analysis. According to measurement category, reliability analysis was most commonly conducted in auditory perceptual studies, with all auditory perceptual papers conducting some form of reliability analysis. Reliability analysis was also common in the acoustic and visual imaging categories, with just under 50% of papers in both categories reporting reliability ratings. While 25% of papers in the aerodynamic category involved reliability analysis, this was least common in the physiologic category, with only one of 11 papers reporting reliability. Sensitivity was reported in one paper, which was common to both the acoustic and visual imaging categories. Specificity analysis was not conducted in any measurement category.

Of the papers which included reliability checking, two performed exclusively intra-rater reliability, while seven solely performed inter-rater reliability analysis. Five papers examined both intra- and inter-rater reliability. For intra-rater reliability, the number of samples re-rated for the purposes of reliability ranged from 10% [24, 35] through to 36% [50], with reliability agreement ranging from an ICC value of 0.341 (one rater with poor intra-rater reliability [24]) to an ICC value of 0.975 [12]. Of those studies examining inter-rater reliability agreement, the number of samples re-rated varied from 10% [35] to 100% [7, 23–25, 36–38]. Inter-rater reliability agreement ranged from an ICC value of 0.145 [24] to 0.998 [54].

The metrics used to assess both intra- and inter-rater reliability included the intraclass correlation coefficient [23–25], Pearson product-moment correlations and absolute difference [12, 49], Pearson's correlation coefficient [35, 54] and Cohen's kappa [25, 33]. Cronbach's alpha [50] was used to determine intra-rater reliability in a single study, while percentage agreement [7, 37, 38], the general linear model and repeated measures of analysis [35] and the Wald 99% confidence interval [34] were used only for inter-rater reliability calculations. It should be noted that percentage agreement, as used in Shiba and Chhetri, Freeman et al., Cooke et al. and Koike's studies should not be used as a standalone statistical measure for inter-rater reliability assessment, as these percentages do not account for concurrence that can be expected by chance, and ultimately does not represent a robust means of determining reliability agreement [25].

Only one of the 35 included studies conducted sensitivity analysis, with no studies conducting an analysis on specificity. Kunduk [35] posed a research question specifically related to

sensitivity, determining whether the timing characteristics, pattern of adduction, start of vocal fold vibration and number of cycles required for the vocal folds to reach full vibration were sensitive to aging, as measured by the VIP. It found that timing characteristics during the VIP were sensitive to the effects of aging, with all timing variables being higher in the older group (mean age 76 years) than the younger group (mean age 26 years). However, the only measure found to reach a significant difference between the younger (mean = 11 cycles) and older groups (mean = 14 cycles) was the number of vocal fold oscillatory cycles before full length vocal fold vibration was achieved (p = 0.001). Across the remaining 34 studies, a select few made a comment relating to sensitivity when interpreting their results [30, 37, 47, 54, 55], however no sensitivity analyses was completed.

While most studies did not report sensitivity nor specificity analysis, 18 of the 35 did seek to use their chosen measure/s of voice onset to differentiate between voice onset types. However, many of these provided an in-text description of what appeared to differ across voice onset types (e.g., how a particular waveform or kymograph varied between breathy and hard onsets), rather than offering numerical cut-off values.

Overall, while the abovementioned papers report reliability outcomes to be of an acceptable level across studies, and VIP to be a sensitive measure of voice onset in detecting age-related differences between patients for the number of vocal fold oscillatory cycles, collectively it is clear that most voice onset measures have not been studied to the level required to be certain of their reliability, sensitivity and specificity.

## GRADE evaluation of research quality

All authors used the GRADE system to evaluate research quality. This evaluation was completed immediately following data extraction for each study. Across all papers, the certainty of evidence as evaluated by GRADE ranged from 'very low' to 'moderate', with 27 of 35 papers falling in the 'very low' category, seven papers classed as 'low' certainty and one as 'moderate'. GRADE certainty assessment values were similarly low across all measurement categories, with the single study assessed as moderate evidence certainty being classed within the 'visual imaging' category.

Acoustic analysis studies ranged from very low to low, with 14 categorised as 'very low' and five as 'low' certainty of evidence. Those four studies exploring aerodynamic analysis were all classed as 'very low' certainty of evidence, as was the case for six of the auditory perceptual papers, with one being classed as 'low' evidence certainty. The eleven physiological papers ranged from 'very low' to 'low' evidence certainty, with eight being 'very low' and three falling in the 'low' certainty of evidence category. Visual imaging was the voice onset measurement category with the largest number of papers, ranging from 'very low' to 'moderate' certainty of evidence. Amongst these papers, 19 were rated as 'very low', three as 'low' and a single paper was deemed to have 'moderate' certainty of evidence.

## Discussion

### Summary of main findings

Across the 35 studies included in this systematic review, all methods of voice onset measurement examined could be classified into one of five categories: auditory perceptual, acoustic, aerodynamic, physiological measures and visual imaging. These studies were evaluated as showing low level of evidence, ranging from very low to moderate certainty of evidence according to the GRADE rating system. Collectively, we found that the reviewed literature presents high variability in vowel onset measures, methodology and automated processes applied, with a lack of robust, high-quality data for any given measure of vowel onset. The

voice onset measure explored by the greatest number of studies was VAT, having been examined in seven studies with the highest quality paper reflecting a GRADE rating of low-quality evidence. The paper with the highest evidence rating according to the GRADE system was of moderate evidence certainty [49], with all other papers being rated as low or very low. Overall, none of the 35 papers in question present high quality research evidence, with a clear paucity of studies examining measures of voice onset in a clinical context. As such, the present literature findings prevent a conclusion of which measures of voice onset would yield the most reliable results with satisfactory sensitivity and specificity to be used in clinical practice.

## Heterogeneity in dataset

The collective data preclude a conclusion pertaining to the most reliable, sensitive and specific measures of voice onset for a variety of reasons. Firstly, across the 35 papers, there is great heterogeneity in the study populations used. There is variability in sample size, ranging from 1 to 112 participants per study and in ages explored, with those studies which report the age of their participants extending from ages eight to 87 years. A further source of variability is the genders included across the studies, with those which report the gender of their participants having an exclusively female or male population or a combination of both. Furthermore, the inclusion of a control or dysphonic group within each paper varies greatly. While most papers only examined normophonic participants, seven involved either an exclusively voice-disordered population or a matched group of participants with voice disorders, with diagnoses ranging from neurological disorders (spasmodic dysphonia) to vocal hyperfunction (vocal nodules) and malignant conditions (laryngeal cancer). Collectively, this extensive scope of participant demographics in each study population prevents both the generalisation of these findings to a larger population and the ability to draw an informed and cohesive conclusion pertaining to the reliability, sensitivity and specificity of the voice onset measures explored.

A further source of heterogeneity across the studies is found in the measurement methods used, with studies exploring either auditory perceptual, acoustic, aerodynamic, physiological or visual imaging-based measurement types, or in 18 studies, a combination of these. Across the 35 studies there are 39 different measures of voice onset used. Even in the case of VAT, the most explored voice onset measure in the dataset, there is variability in how this measure is collected, with a difference in approach evident across research groups. This variance in measurement methods over time can be attributed to technological advances. Many vowel-initial measures of voice onset may never reach the stage of becoming clinically practicable as new measures, based on updated technology and approaches, are constantly being developed before existing measures are sufficiently researched and applied to clinical contexts of voice assessment.

The automation of processes throughout the methodology of studies introduces a further source of variation in the voice onset literature. Automation is applied throughout the dataset in the stages of data processing, data generation or a combination of the two, with 27 of the 35 studies using automation in some capacity throughout their methodology. With a vast variety of algorithms and software platforms employed across these studies and the differing stages where these automated processes are applied, it is evident that automation introduces furthermore heterogeneity of measurement across the vowel-initial voice onset literature.

There are several potential sources of this heterogeneity. Voice onset is a complicated measure, such that currently there appears to be no single measure able to quantify it satisfactorily. This may have led to 'exploratory' studies in the absence of a theoretical model of voice onset, which introduces variation in the way vowel onset is measured and explored. Other sources can be attributed to the array of robust research indicators which are presently lacking across

the vowel onset evidence base. The current evidence lacks well-designed studies which include a pre-calculated sample size, random sampling of the study population, theoretical models and reliability, sensitivity and specificity ratings for outcome measures of interest, reasonable rationales for vocal tasks used, voice disorder classification criteria, focal voice disorder populations (i.e., currently there are mixed population groups, such as functional and organic voice disorder types) and standardised voice onset measurement protocols. This range of factors can likely be attributed to the extensive variation between each of the studies which make up the collective set.

This heterogeneity in turn, limits interpretation and generalisability of the presented data. Across the study set, limited and underestimated sample sizes are highly prevalent, with all studies lacking a pre-calculated sample size with sufficient statistical power. This limits the ability to meaningfully interpret any data and apply this to larger populations. The lack of standardised protocols and reliability analyses across the reviewed studies is another contributing factor, which results in issues with the data reported and difficulty in interpreting this. Finally, the inconsistencies in methodology, outcome measures, measurement techniques and results across studies make it exceedingly difficult to draw significant trends and conclusions.

Collectively, there is great variability in the measurement of the voice onset phenomenon from methodological approach through to selected voice onset measure, leading to a vast array of data that can't easily be replicated, interpreted nor synthesised. This heterogeneity prevents us from ascertaining the clinical utility of each respective measure and as such, disallows us from forming any generalisations pertaining to clinically valid measures of vowel onset. The diversity in methods and approaches highlights the lack of a commonly accepted standard when performing voice onset analysis, which further limits the opportunity to appreciate how voice onset could best, most reliably, sensitively and specifically be applied in a clinical context.

## Voice onset definitions

An added limitation of the study findings is grounded in the lack of accepted definitions pertaining to voice onset in vowel-initial contexts. While most studies provided some form of voice onset definition, there was considerable variation between these; with ten defining only the specific voice onset measure/s examined in their study and a further ten papers describing voice onset according to a clear and detailed definition which accounted for the range of physiological processes involved. Of the papers which did not specify the meaning of voice onset, these often reported providing instruction, training and/or modelling to study participants which is not detailed in each paper (for example, [36]). Training of subjects requires perceptual judgement of voice onset by trainers and speakers to perform the voice onset. Therefore, the lack of independent verification of perceptual features present in the samples where auditory perceptual ratings were not used is problematic. This lack of reporting also limits the opportunity for replicability and consistency between studies. Without the provision of clear and explicit definitions of vowel-initial voice onset across the literature, it is difficult to establish if the phenomenon being measured is in fact voice onset. Given that the definition of voice onset informs the methodology and nature of research conducted across each study, this discrepancy across the collective dataset is a clear contributing factor to the heterogeneity of study design and outcomes.

The issue of ambiguity surrounding what specifically is being measured as voice onset is further compounded by the lack of correlation with auditory perceptual judgements throughout the collective group. With only three of the papers correlating their instrumental measures of voice onset with a perceptual judgement of onset type, most papers are neglecting the gold

standard of voice assessment and in so doing, bringing into question the validity of their chosen measures of vowel-initial voice onset.

## Quality of evidence

The GRADE findings of this review evidenced that the quality of papers throughout the vowel-initial voice onset literature is low, informed largely by the research design and small sample size of all studies examined. Amongst these papers there was a low incidence of reliability assessments to ascertain the reproducibility of research findings, with some form of reliability assessment occurring in only 14 of the 35 papers. Across these papers, these ratings tended to be quite variable, including instances of low reliability reported. This may have resulted from factors pertaining to the raters themselves (i.e., variation in clinical experience, skill set and training in use of the measurement tool) but is most likely attributable to elements associated with research quality, such as study design, sample size and sampling methods. A cross-sectional study is typically less reliable than prospective or cohort studies, small sample sizes yield less reliable results than studies involving greater participant numbers and convenience sampling is generally less reliable than random sampling. With cross-sectional studies being the most common study design and the use of small sample sizes attained through convenience sampling across the 35 papers, the overall low quality of the collective paper set elucidates some causative factors behind the low and variable reliability results reported in this review.

Compared to reliability analysis, even lower rates of sensitivity assessment were performed with only a single study reporting some form of sensitivity analysis, and nil studies were found to analyse specificity. Almost none of the reviewed studies used voice onset measures to discriminate disordered from non-disordered speakers. Furthermore, voice onset measures were not used as an outcome to detect participants' vocal condition. These factors help to account for the lack of discrimination analyses conducted across the studies.

## Strengths and limitations

The papers included in the systematic review covered all types of relevant literature available at the time of the study, featuring a comprehensive search strategy including both published papers and grey literature sources. Updated searches were conducted in December 2022 and May 2023 to ensure all recently published articles of interest were considered for review. Limitations of the study approach include only examining literature published in the English language i.e., excluding non-English sources, and not performing a further citation search of the two studies added to the dataset from the final updated literature search, which may have potentially sourced further studies of relevance. A lack of quantitative data and a high level of heterogeneity between the studies prevented the conduction of a quantitative analysis of the collective study findings. The dearth of data conducted beyond a laboratory-based setting also made it difficult to determine which measures of voice onset may be most practical for application in clinical contexts. As such, we are unable to develop well-informed recommendations and conclusions pertaining to how voice onset may be most effectively measured in patient scenarios, as these conclusions would not be supported by research we would describe as reliable, sensitive nor specific.

## Comparison with other studies

Nil other review studies have been conducted into vowel-initial voice onset measurement to enable a direct comparison with the existing literature, however, several studies have recognised that the existing pool of voice onset measurement literature presents a heterogeneous set of data and low level of evidence methodologies. For example, Patel [49] reported that studies

investigating the onset of phonation examine small cohorts of vocally healthy adults and have utilised different waveform types, which yields variable findings. Likewise, Petermann and colleagues [62] recognised that the present literature involves different approaches to measuring even the same voice onset measure, with no standardised processes in place and wide inter- and intrasubject variability, which complicates the cross-study comparison of results. Maryn and Poncelet [24] also recognised the failings of the existing voice onset literature in examining or developing a range of quantitative, objective voice onset measures, without any application to clinical voice assessment protocols nor patient-centric contexts.

## Clinical implications

The lack of an accepted standard pertaining to vowel-initial voice onset measurement in clinical contexts is directly evidenced in the range of clinical voice assessment proformas which lack an assessment of this feature. Despite the utility of vowel-initial voice onset in providing predictive information pertaining to the voice function that follows, the plethora of studies relating to vowel-initial voice onset measures have proved trivial in bridging the gap between theory and practice; failing to identify a single form of measurement which is proven to yield reliable, sensitive and specific results which can be applied to clinical voice patient contexts. Until such a measurement tool can be identified and researched to prove its utility as a clinically valid measure, it seems that clinical voice assessment and the standardisation of voice assessment tasks will continue to be limited by the current gaps in the voice onset literature.

## Implications for research and future studies

Further, high quality research is clearly needed in the vowel-initial voice onset measurement space, preferably, within the next five to ten years. These papers would ideally involve a comparison of voice onset measures using methods of assessment which could easily and efficiently be applied in clinical contexts, as well as validation of these individual measures. In addition, further research into standardised measurement criteria and voice assessment protocols which incorporate clinically viable measures of vowel initial voice onset would prove valuable. Given that vowel-initial voice onset measures provide useful information for all voice disorder populations, diverse populations and disorder types would need to be considered. Performing effect size calculations which are clearly documented in the resulting manuscript, and seeking large study populations wherever possible should be prioritised.

Further research should also perform independent auditory perceptual ratings of samples for cross-comparison; ideally using publicly available voice databases wherever possible. It is also of utmost importance that future voice onset research presents a physiological definition of what precisely each study will measure, rather than measuring voice onset solely according to perceptual judgements of voice onset type. In the same vein, these studies must also ensure that the measure they select is able to assess these physiological features, rather than base a measurement upon inference. The development of such research would lead to far greater confidence in the collective findings across the vowel-initial voice onset literature, and an ability to develop informed recommendations pertaining to the application of these measures in a clinical capacity.

## Conclusion

Voice onset is a highly variable event involving multiple physiological processes and as such, is a difficult phenomenon to measure. The findings of this review do not permit us to provide informed recommendations regarding the most reliable, sensitive and specific means of measuring vowel-initial voice onset, due to the heterogeneity and overall low research quality of

the examined studies. There is a clear need for high-quality data and well-designed research which examines voicing control across the lifespan and across disorders. Ideally, this should compare a range of measures, particularly those which would be easily practicable in clinical scenarios, and provide a robust evaluation of their reliability, sensitivity and specificity in patient-based contexts.

## Supporting information

**S1 Checklist. PRISMA 2020 checklist.**
(DOCX)

**S1 File. Systematic review protocol.**
(DOCX)

**S2 File. Database search strategy.**
(DOCX)

**S3 File. Data extraction summary table.**
(DOCX)

## Author Contributions

**Conceptualization:** Antonia Margarita Chacon, Catherine Jeanette Madill.

**Data curation:** Antonia Margarita Chacon, Duy Duong Nguyen, John Holik, Michael Döllinger, Tomás Arias-Vergara.

**Formal analysis:** Antonia Margarita Chacon, Duy Duong Nguyen, John Holik, Catherine Jeanette Madill.

**Investigation:** Antonia Margarita Chacon, Duy Duong Nguyen, Catherine Jeanette Madill.

**Methodology:** Antonia Margarita Chacon.

**Project administration:** Antonia Margarita Chacon.

**Resources:** Antonia Margarita Chacon.

**Validation:** Antonia Margarita Chacon.

**Visualization:** Antonia Margarita Chacon.

**Writing – original draft:** Antonia Margarita Chacon.

**Writing – review & editing:** Antonia Margarita Chacon, Duy Duong Nguyen, John Holik, Michael Döllinger, Catherine Jeanette Madill.

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
