## [Decision Letter · Decision Letter 0]

17 Jan 2024

PONE-D-23-27828Vowel onset measures and their reliability, sensitivity and specificity: A systematic literature reviewPLOS ONE

Dear Dr. Chacon,

Thank you for submitting your manuscript to PLOS ONE. After careful consideration, we feel that it has merit but does not fully meet PLOS ONE’s publication criteria as it currently stands. Therefore, we invite you to submit a revised version of the manuscript that addresses the points raised during the review process.

We look forward to receiving your revised manuscript.

Kind regards,

Antonino Maniaci

Academic Editor

PLOS ONE

Journal Requirements:

Did you know that depositing data in a repository is associated with up to a 25% citation advantage (https://doi.org/10.1371/journal.pone.0230416)? If you’ve not already done so, consider depositing your raw data in a repository to ensure your work is read, appreciated and cited by the largest possible audience. You’ll also earn an Accessible Data icon on your published paper if you deposit your data in any participating repository (https://plos.org/open-science/open-data/#accessible-data).

"Contributions by Michael Döllinger were supported by Deutsche Forschungsgemeinschaft (DFG) under grant no. DO1247/8-2. Contributions by Tomás Arias-Vergara were supported by Deutsche Forschungsgemeinschaft (DFG) under grant no. SCHU3441/3-2. Contributions by Antonia Margarita Chacon and Duy Duong Nguyen were supported by the Doctor Liang Voice Program at The University of Sydney."

Funding information should not appear in the Acknowledgments section or other areas of your manuscript. We will only publish funding information present in the Funding Statement section of the online submission form. 

"Contributions by MD were supported by Deutsche Forschungsgemeinschaft (DFG) under grant no. DO1247/8-2. Contributions by TA were supported by Deutsche Forschungsgemeinschaft (DFG) under grant no. SCHU3441/3-2. Contributions by AC and DN were supported by the Doctor Liang Voice Program at The University of Sydney. The funders had no role in study design, data collection and analysis, decision to publish, nor preparation of the manuscript."

4. We note that you have referenced (ie. unpublished dissertations/works) which has currently not yet been accepted for publication. Please remove this from your References and amend this to state in the body of your manuscript: (ie “Bewick et al. [Unpublished]”) as detailed online in our guide for authors

5. We note you have included a table to which you do not refer in the text of your manuscript. Please ensure that you refer to Tables 3, 5 and 6 in your text; if accepted, production will need this reference to link the reader to the Table.

6. We note that this manuscript is a systematic review or meta-analysis; Please upload copies of the completed PRISMA checklist as 'Supporting Information' with a file name “PRISMA checklist”.

**Additional Editor Comments:**

Dear authors, perform minor revisions required.

Bests

Reviewers' comments:

Reviewer's Responses to Questions

**Comments to the Author**

1. Is the manuscript technically sound, and do the data support the conclusions?

Reviewer #1: Yes

Reviewer #2: Partly

2. Has the statistical analysis been performed appropriately and rigorously? 

Reviewer #1: Yes

Reviewer #2: Yes

3. Have the authors made all data underlying the findings in their manuscript fully available?

Reviewer #1: Yes

Reviewer #2: Yes

4. Is the manuscript presented in an intelligible fashion and written in standard English?

Reviewer #1: Yes

Reviewer #2: Yes

5. Review Comments to the Author

Reviewer #1: Well written manuscript. Well done to all authors. I would just like to highlight minor comments.

Methods

Line 118- I would suggest to put MEDLINE via OVID, EMBASE via OVID

Study population characteristics

Line 204- I would suggest putting these summarized details in a table 1, it would be easier to read. Don’t have to repeat them in text.

Line 518- above mentioned.

Supplement materials

Please include search strategy for all the databases.

Reviewer #2: Abstract

- Purpose: To systematically review validity of vowel onset measures for clinical voice assessment.

- Methods: Electronic databases searched for studies measuring vowel onset using >1 instrument. Data extracted on measures, reliability, sensitivity, specificity.

- Results: 35 studies covered 5 measurement categories; 39 measures identified. Measures showed low-to-moderate reliability; sensitivity rarely assessed.

- Conclusions: Heterogeneity in populations and methods preclude conclusions on most valid measures. Standardization of research methodology is needed.

Introduction

- Note prevalence of voice disorders and importance of early/accurate diagnosis

- Outline current clinical assessment methods and limitations (perceptual only)

- Define voice onset and importance in assessment/treatment monitoring

- Highlight variability in instrumental onset measures used across studies

- State importance of establishing valid clinical measures for standardized assessment

- Explicitly state research questions:

- What vowel onset measures have been used instrumentally?

- What is the evidence for their reliability, sensitivity and specificity?

- Hypothesize most effective measures will demonstrate high reliability, sensitivity, specificity

- Aim to inform standardized assessment by identifying valid instrumental onset measures

Methods

- Specify retrospective systematic review design

- Provide inclusion/exclusion criteria in detail

- Describe all information sources and search strategy used

- Explain screening, data extraction and analysis processes

- Detail all variables/data extracted from studies

- Describe approach to risk of bias/quality assessment

- State programs/statistical analyses used

Results

- Report study selection details quantitatively in PRISMA flow diagram

- Provide characteristics of included studies as evidence tables

- Categorize measures used and define each clearly

- Present findings for each measurement category separately:

- Report measures used, populations, reliability, etc. numerically

- Include relevant tables/figures to organize data

- Note any similarities/differences between categories

- Acknowledge limitations like heterogeneity between studies

Discussion

- Compare findings to other reviews on this topic and highlight novel insights

- Critically discuss sources of heterogeneity between studies (measures, populations, etc.)

- Analyze how heterogeneity limits interpretation and generalizability

- Explore possible reasons for low/variable reliability reported

- Address lack of sensitivity/specificity analyses between studies

- Relate findings to current clinical practices and limitations

- Recommend standardized measurement criteria/protocols for future research

- Suggest priorities for future studies based on gaps identified

- E.g. Validating individual measures, diverse populations/disorders, automation

- Discuss implications of findings for voice assessment standardization

- Conclude by emphasizing need for high-quality data before recommending measures

- Acknowledge limitations like excluding non-English sources

- Consider strengths like comprehensive search and inclusion of gray literature

6. PLOS authors have the option to publish the peer review history of their article (what does this mean?). If published, this will include your full peer review and any attached files.

Reviewer #1: No

Reviewer #2: **Yes: **Salvatore

---

## [Author Response · Author response to Decision Letter 0]

25 Feb 2024

Please see 'Response to Reviewers' (uploaded to submission system) for responses to all reviewer and editor comments.

---

## [Decision Letter · Decision Letter 1]

24 Mar 2024

Vowel onset measures and their reliability, sensitivity and specificity: A systematic literature review

PONE-D-23-27828R1

Dear Dr. Chacon,

We’re pleased to inform you that your manuscript has been judged scientifically suitable for publication and will be formally accepted for publication once it meets all outstanding technical requirements.

Kind regards,

Li-Hsin Ning

Academic Editor

PLOS ONE

Additional Editor Comments (optional):

Reviewers' comments:

Reviewer's Responses to Questions

**Comments to the Author**

1. If the authors have adequately addressed your comments raised in a previous round of review and you feel that this manuscript is now acceptable for publication, you may indicate that here to bypass the “Comments to the Author” section, enter your conflict of interest statement in the “Confidential to Editor” section, and submit your "Accept" recommendation.

Reviewer #1: All comments have been addressed

Reviewer #2: All comments have been addressed

2. Is the manuscript technically sound, and do the data support the conclusions?

Reviewer #1: Yes

Reviewer #2: Yes

3. Has the statistical analysis been performed appropriately and rigorously? 

Reviewer #1: Yes

Reviewer #2: Yes

4. Have the authors made all data underlying the findings in their manuscript fully available?

Reviewer #1: Yes

Reviewer #2: Yes

5. Is the manuscript presented in an intelligible fashion and written in standard English?

Reviewer #1: Yes

Reviewer #2: Yes

6. Review Comments to the Author

Reviewer #1: (No Response)

Reviewer #2: Both research ethics and publication ethics are essential for maintaining the integrity of the scientific community and ensuring that research findings are reliable, trustworthy, and accountable. They provide a framework for researchers, institutions, and publishers to uphold ethical standards and promote responsible conduct in research and scholarly publishing. Violations of research ethics and publication ethics can have serious consequences, including damage to reputation, loss of trust, and potential harm to individuals or the scientific community as a whole.

7. PLOS authors have the option to publish the peer review history of their article (what does this mean?). If published, this will include your full peer review and any attached files.

Reviewer #1: **Yes: **RANITA HISHAM SHUNMUGAM

Reviewer #2: **Yes: **Salvatore Lavalle

---

## [Editor Report · Acceptance letter]

1 Apr 2024

PONE-D-23-27828R1 

PLOS ONE

Dear Dr. Chacon, 

I'm pleased to inform you that your manuscript has been deemed suitable for publication in PLOS ONE. Congratulations! Your manuscript is now being handed over to our production team.

Kind regards, 

on behalf of

Dr. Li-Hsin Ning 

Academic Editor

PLOS ONE